



# Upwind vs downwind: Loads and acoustics of a 1.5 MW wind turbine

Pietro Bortolotti, Lee Jay Fingersh, Nicholas Hamilton, Arlinda Huskey, Chris Ivanov, Mark Iverson, Jonathan Keller, Scott Lambert, Jason Roadman, Derek Slaughter, Syhoune Thao, and Consuelo Wells

National Renewable Energy Laboratory, 15013 Denver West Parkway, Golden, CO 80401, USA

**Correspondence:** Pietro Bortolotti (pietro.bortolotti@nrel.gov)

**Abstract.** This paper discusses the motivation, preparation, risk mitigation, execution, and results of a full-scale experiment where the rotor of a 1.5 MW wind turbine was operated in a downwind configuration. The experiment took place at the National Renewable Energy Laboratory Flatirons Campus in Colorado, USA, and involved the collection of loads and power together with acoustic measurements from an array of four microphones. 410 min of downwind operation and 960 min of conventional

upwind operations are used to validate the numerical predictions of the aeroelastic solver OpenFAST in terms of loads and performance. In the wind speed range from 4.5 to 12.5 m s$^{-1}$ the downwind rotor generates higher damage equivalent loads for the blade root flapwise moment, blade root edgewise moment, and tower-base fore-aft moment compared to the upwind rotor. For these metrics of fatigue loads, numerical predictions match well the experimental observations. OpenFAST is however also seen underpredicting a power gain in the downwind rotor. In terms of acoustics, the overall sound pressure levels recorded in the

field are similar between the upwind and downwind cases, but downwind operation worsens the metrics describing amplitude modulation. The paper closes with the recommendation to further investigate the potential of downwind rotor technology for floating wind applications, where the tilt angle of downwind rotors can compensate for the pitching of the floating platform.

## 1 Background

The vast majority of modern multi-megawatt wind turbines mount an upwind rotor, i.e., the rotor faces the wind, and the blades spin in front of the turbine tower. Upwind-oriented wind turbines have, however, not always dominated the utility-scale market. Multiple early prototypes in the 1980s flew downwind rotors (Manwell et al., 2010), and a decade ago, the manufacturer Hitachi designed, installed, and operated 2 MW and 5 MW downwind-oriented wind turbines in Japan. The designs and installations were both land-based and offshore. Also, downwind rotors are still fairly common in distributed wind applications, and some

new multi-megawatt concepts for floating applications mount downwind rotors [1] [2]. Commercial multi-megawatt wind turbines

---

[1]https://www.x1wind.com

[2]https://www.windpowermonthly.com/article/1880758/mingyang-completes-166mw-oceanx-twin-rotor-floating-offshore-wind-platform





mount upwind rotors because of historical challenges experienced by downwind rotors linked to increased acoustic emissions and fatigue loads. Both effects are caused by the interaction of the blades with the wake of the tower. In the 1980s, the two-bladed downwind rotor of the MOD-1 wind turbine installed near Boone, North Carolina, was a nuisance to residents due to excessive acoustic impulses. An analysis led by the Solar Energy Research Institute showed that the source of this excessive

acoustic emission was the dynamic behavior of the lift force generated by the blades as they passed behind the lattice tower. The acoustic emissions were found to be influenced by both the complex terrain where the turbine was installed and by the various atmospheric conditions in which the MOD-1 turbine was operating (Kelley et al., 1985). The negative impact of the MOD-1 wind turbine on the neighboring communities contributed to the dominance of upwind rotors since then.

## 1.1  Why downwind?

Wind turbines continue to grow in rotor size, hub height, and nameplate power. The race for the biggest wind turbine is clearly visible in the offshore market, where manufacturers have been breaking records almost yearly. In the land-based market, although growth is present, the trend has taken a different path. Wind turbines with nameplate power above 5 MW appeared more than a decade ago but have been winning a sizeable share of the market only recently. This slowdown in the growth of average ratings can be explained by the push to reduce specific power (the ratio of turbine rating to rotor swept area), as

discussed in Bolinger et al. (2020). The trend of decreasing specific power is now challenged by the arrival of cost-competitive storage systems such as batteries and hydrogen and might reverse in the coming years (Wiser et al., 2022). Nonetheless, the growth in rotor size and hub height is expected to continue, pending technology innovations. Increasingly larger upwind rotors face the growing challenge of complying with the clearance between blades and tower, a constraint that downwind rotors help to alleviate. In this scenario, downwind rotor configurations resurfaced as a promising option for the next generation of

machines.

The first phase of the Big Adaptive Rotor (BAR) project discussed the value of highly flexible 100 m wind turbine blades, which offer a pathway to reduce capital costs and levelized cost of energy, as well as to alleviate logistics constraints (Bortolotti et al., 2021). The study showed the technical viability of transporting these blades by using controlled flexing during rail transport. There may be economic benefits to the controlled flexing solution compared to adopting spanwise segmentation.

To avoid derailing train flatcars during horizontal curves, the blades must be more flexible than usual. To accommodate the additional flexibility, the blades could be mounted either in downwind rotor configurations or in highly tilted upwind rotors. Taking advantage of the additional tilt of downwind turbines for farm-level flow benefits has also shown promising results in the numerical analyses presented in Annoni et al. (2017); Bay et al. (2019); Cossu (2021b, a) and scaled wind tunnel experiments (Scott et al., 2020; Nanos et al., 2020). Numerical and experimental studies show a power improvement of the

wind farm when wind is aligned with a row of wind turbines whose rotors are tilted in a downwind configuration. Tilted downwind rotors deflect wakes downwards as opposed to upwards, and increased vertical entrainment allows downstream turbines to increase their power output. At lower fidelity, Bay et al. (2019) report advantages between 1.5 % and 2 % in annual energy production by tilting the rotors of the turbines located on the perimeter of a 45-turbine wind farm with a 7x7 grid, leaving the four corner grid positions empty. At higher fidelity, Cossu (2021b) presents results for a 3x6 wind farm, where six





turbines experience undisturbed wind speed and twelve turbines sit in the full wake. The results were generated by running 3D computational fluid dynamics with an actuator disk model. Power gains up to +40 % were obtained thanks to a combination of high tilt angles (30°) of the front and middle rows of wind turbines and a higher axial induction of the tilted rotors. Within BAR, Frontin et al. (2024) extended the high-fidelity analysis to a full sweep of wind directions for a 4x4 wind farm. These new results were far less promising, with an overall reduction in annual energy production. It should be noted that Frontin et al. (2024) explored only one wind farm configuration and did not explore the dependency of power performance on key parameters such as farm spacing or atmospheric stability.

A number of publications have also investigated the potential of highly coned downwind rotors. Conceptual studies (Loth et al., 2017) were followed by more detailed studies (Pao et al., 2021) and field tests (Kaminski et al., 2023b, a). Highly coned rotors offer a pathway to lower blade flapwise loads and consequently mass and cost savings. A last area where downwind rotors could be advantageous over upwind rotors is floating applications, where the annual energy production loss caused by the reduced swept area due to the platform pitch angle could be balanced by the downwind nacelle tilt angle. Overall, the techno-economic viability of downwind rotors has been under investigation for decades, with alternating conclusions (Bortolotti et al., 2022).

## 1.2 Goal of the experiment

In this scenario, the fatigue loads and acoustic emissions of large downwind rotors remain an open research topic. To narrow this gap, the BAR research team at the National Renewable Energy Laboratory (NREL) designed, planned, and executed an ambitious experiment with the goal of creating a dataset that can be used to validate numerical predictions in terms of loads, performance, and aeroacoustics of upwind and downwind rotors. The experiment aims to support the investigations around the feasibility, reliability, performance, and economic viability of downwind-oriented wind turbines. The experiment consists of operating a 1.5 MW wind turbine located at the NREL Flatirons Campus in a downwind configuration while monitoring loads, power, and aeroacoustic emissions, which can then be compared to the upwind operation of the wind turbine. To switch from upwind to downwind, the nacelle of the turbine has to yaw by 180°, and the three blades have to pitch by 180°. For the same wind and the same observer, the rotor keeps spinning clockwise when viewed from upwind. The rotor, however, spins in the opposite direction with respect to the nacelle. The pitch rate–both speed and direction–remains the same, but the pitch actuators need to operate between 180° and 270°. The unique dataset that was generated during the experiment is used to validate the numerical predictions in terms of dynamic behavior and performance for both upwind and downwind rotor configurations. The predictions are generated by the aeroservoelastic solver OpenFAST. The experimental aeroacoustic emissions are also compared between upwind and downwind in terms of overall sound pressure and sound power levels as well as amplitude modulation.

## 1.3 Structure of the paper

The next sections describe the design, planning, execution, and results of the experiment. The test turbine and the test site are introduced in Section 2. Next, Section 3 discusses the approach followed to ensure the safety and success of the experimental



campaign, whereas Section 4 discusses the data collection and data analysis. Results are discussed in Section 5, with loads presented in Section 5.1 and acoustics in Section 5.2. The conclusions of the paper are given in Section 6.

## 2   Test turbine and test site

Loads and aeroacoustic measurements were taken at and around a 1.5 MW wind turbine manufactured and sold by GE Vernova. This model is representative of a large segment of the installed wind capacity, with more than 18,000 turbines of this make and model currently in operation in the United States. The GE 1.5 MW wind turbine is built on the platform of GE 1.5 MW SLE commercial wind turbine model and was installed at the NREL Flatirons Campus in 2008 with the objective of supporting wind energy research initiatives. The turbine has a horizontal axis and is a three-bladed, upwind turbine with full span pitch control. Table 1 provides the key descriptive information of the test turbine.

**Table 1.** Main characteristics of the test turbine.

| | |
|---|---|
| Turbine manufacturer and address | GE Vernova, 300 Garlington Rd., P.O. Box 648, Greenville, SC 29602-0648 |
| Model | GE 1.5 MW SLE |
| Rated power (kW) | 1500 |
| Rated wind speed (m s$^{-1}$) | 14 |
| Serial number | N000780-N / TB059-3 |
| Blade make, type, and serial number | GE37c, made of fiberglass |
| Generator make, type, and serial number | Winergy, doubly fed induction, JFEC-500SS-06A |
| Gearbox make, type, and serial number | Winergy multistage planetary/helical model PEAB 4410.4 |
| Control software | WindSCADA |
| Wind turbine type | Horizontal axis, upwind |
| Tower type | Tubular |
| Number of blades | 3 |
| Hub height (m) | 80 |
| Rotor diameter (m) | 77 |
| Horizontal distance from rotor center to tower axis (m) | 3.8 |
| Speed control | Pitch control |
| Constant or variable speed | Variable |
| Rated rotor speed (rpm) | 18.3 |

The NREL Flatirons Campus is located 13 km south of Boulder, Colorado, and the turbine is located on test site 4.0. Figure 1 shows the test turbine with its rotor in an idling downwind configuration (the leading edges of the three blades point toward the tower). The acoustic instrumentation was arrayed in the area downstream of the turbine along the prevailing wind direction, which, at the NREL Flatirons Campus, is 285°. During the experiment, the team made sure to minimize sources of background noise. The NREL Flatirons Campus has other wind turbines installed, but none was operated at the time of the experiment.





The campus also had ongoing construction, and especially noisy activities were postponed during the collection of acoustic data. A proprietary aeroelastic model of the turbine is available in NREL's OpenFAST framework. In the aeroelastic model, the industrial controller in the form of a compiled dynamic link library drives the generator torque and the pitch actuation.

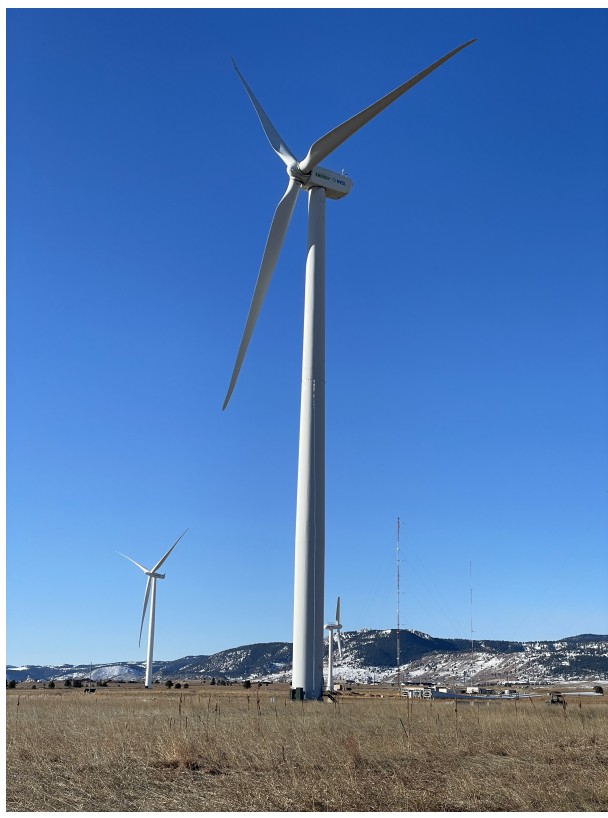

**Figure 1.** GE 1.5 MW wind turbine located on site 4.0 at the Flatirons Campus, with the rotor in an idling downwind configuration. Photo credit: Rafael Mudafort, NREL.

## 3 Planning and execution of the experiment

The experiment followed the approaches used in previous campaigns to characterize the mechanical loading of the turbine in various operating conditions (Santos and van Dam, 2015) and the acoustic emissions under yaw offsets (Hamilton et al., 2021). The experiment included standard load measurements along the turbine components following the International Electrotechnical Commission (IEC) standard, namely IEC 61400-13 (2015), combined with multiple point measurements located to sample the acoustic emissions, including points required and suggested in IEC 61400-11 (2018).

Dozens of channels of data were acquired from instrumentation installed on the 135 m-high M5 meteorological tower, which is located approximately two rotor diameters upstream of the test turbine, along with turbine supervisory control and data acquisition (SCADA) system channels. These include:





- – Flapwise and edgewise moments at the blade root

– Fore-aft, side-side, and torsional moments at tower-top

- – Fore-aft and side-side moments at the tower-base

- – Main shaft torque and bending moment

- – Generator power, generator torque, rotor speed, pitch angle, and blade azimuth.

In addition, acoustic levels were recorded from four microphones located in the field. Acoustic data collection followed the
120 procedure of IEC 61400-11 (2018), with the inclusion of additional microphones for directionality. One primary microphone
was located at the IEC location, and three additional low-frequency microphones were placed 157-m away from the turbine.
Figure 2 shows the locations of the met tower and microphones with respect to the turbine and the prevailing wind direction.

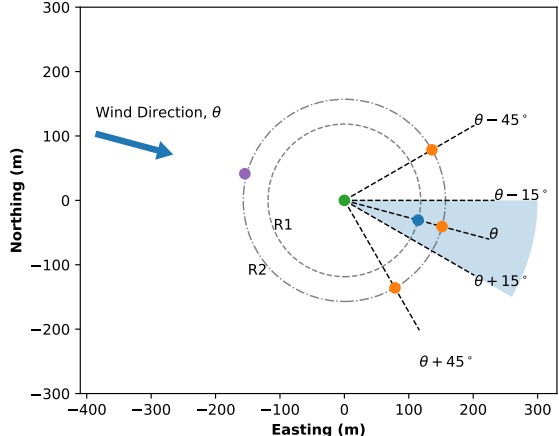

**Figure 2.** Locations of acoustic measurements. R1 is the IEC-prescribed distance, i.e., the turbine height, which is the tower height (80 m)
plus the rotor radius (38.5 m), i.e., 118.5 m (IEC 61400-11, 2018). R2 is R1 plus the rotor radius, i.e., 157 m. The blue dot is at the IEC
location and was the primary measurement location in the audible range. Three low-frequency microphones were installed at the orange dots.
The 135 m-high M5 meteorological tower is located at the purple dot.

Tables A1, A2, and A3 list the instrumentation that were used for the test. Table A1 lists the nonload channels, Table A2 lists
the load channels, and Table A3 lists the aeroacoustics channels. Following the recommendation of IEC 61400-11 (2018), the
125 wind speed was measured downstream of the turbine using a cup anemometer located 10 m above the ground. Microphones
with wind screens were placed in the downwind reference position and optional positions to measure the total and background
noise along with the directivity. All instrumentation was recalibrated ahead of the data collection.



## 3.1 Risk assessment and risk mitigation

Turning the rotor of the GE 1.5 MW wind turbine from upwind to downwind operation was a challenging task, and the
130 experiment was prepared by conducting a failure mode and effects analysis (FMEA) to investigate the risks connected to
the experiment and identify risk mitigation strategies. The FMEA also underwent a third-party review led by Gulf Wind
Technology. In addition to the FMEA, personnel, equipment, and environmental safety reviews were also completed.

### 3.1.1 FMEA

The FMEA was prepared as much as possible according to the guidelines provided by the IEC 60812-2 (2006) standards. The
135 FMEA identified 20 risks, and each risk was ranked with a score for frequency and a score for severity. The product of the
two scores was taken as the risk priority number (RPN). Frequency and severity scores were ranked between 1 and 5, and the
risk severity number could therefore vary between 1 and 25. Table A4 lists the ten risks that scored an RPN of 10 or above.
To mitigate all risks, first, a detailed load analysis was conducted in OpenFAST running design load cases (DLCs) prescribed
by IEC 61400-1 (2019) and load cases that simulate conditions observed at the NREL Flatirons Campus. For the latter set
of simulations, inflow data were extracted from Hamilton and Debnath (2019). Load cases were run at extreme turbulence
intensity and extreme positive and negative shear. Loads were monitored to ensure if and how much they increased due to the
downwind orientation of the rotor; see Section 3.1.2 for more details. In addition, the FMEA identified reversed aerodynamic
thrust as a high-RPN risk. The team defined a mitigation strategy for this risk, which is described in more detail in Section 3.1.3.
Lastly, the FMEA ranked high the risks linked to the smaller blade-to-tower clearance. This risk is discussed in Section 3.1.4.

### 145 3.1.2 Load analysis in OpenFAST

The main element of the de-risking strategy consisted of running a detailed load analysis comparing loads experienced by the
wind turbine in the upwind and downwind orientations. The load analysis was conducted in NREL's open-source framework
OpenFAST v3.3.0. The input files to OpenFAST describing the aeroelastic behavior of the wind turbine were generated from
a legacy Fast v7 model, which was in turn generated from input files of the Flex solver used by GE. Note that blades were
150 modeled as straight Euler–Bernoulli beams, and the effects of prebend were ignored in the structural dynamics of the blades.
Prebend was, however, accounted for in the calculation of the blade–tower clearance.

OpenFAST differentiated between the upwind and downwind configurations by means of the sign used for the following
quantities:

– Location of the center of gravity of the hub, which was set negative for downwind and positive for upwind

– Overhang distance, which was set positive for downwind and negative for upwind

– Shaft location, which was set negative for downwind and positive for upwind

– Shaft tilt, which was set positive for downwind and negative for upwind



– Location of the center of gravity of the nacelle, which was set negative for downwind and positive for upwind.

In the downwind case, the tower shadow was modeled using the model formulated by Powles (1983) and implemented in
OpenFAST by Moriarty and Hansen (2005). The aerodynamic performance of the blades of both upwind and downwind rotors
was simulated modeling lift, drag, and moment curves corresponding to medium-rough airfoil conditions. These airfoil polars
were provided by GE and were validated in Madsen et al. (In Preparation).

The inputs to OpenFAST were validated by comparing the component masses listed in the input files with the component
masses listed in the technical documentation provided by GE at the time of installation and commissioning of the wind turbine.
The numerical predictions and experimental measurements in terms of natural frequencies of the blade and of the entire
system were also compared. The comparison was performed at 0 rpm in both OpenFAST and in the field. The latter set was
extracted after a controlled shutdown maneuver brought the rotor to a full stop. Lastly, shaft tilt, rotor overhang, and tower
clearance at the blade tip were compared between OpenFAST and a 3D computer-aided design model that was generated with
a 3D scanner. Table 2 shows the relative differences. The OpenFAST model shows masses smaller than the specifications.
However, the masses listed in the technical documentation include the fixture used for shipping, and the differences were
deemed acceptable. The natural frequencies predicted by OpenFAST were close to the experimental values listed in Santos
and van Dam (2015). Lastly, differences between OpenFAST and the model generated with the 3D scanner can be seen in the
shaft tilt and clearance at the blade tip. Assuming that the 3D model is accurate, some of the differences can be explained by
the gravitational moment acting on the rotor and effectively reducing the nominal tilt of the wind turbine. An additional test
was conducted by measuring shaft tilt at multiple locations inside the nacelle and at four yaw angles. The measurements show
a range of 0.5° and a dependency on the yaw angle, which might suggest that the tower is itself not perfectly vertical.

**Table 2.** Results of the validation of the OpenFAST model of the GE 1.5 MW wind turbine. Negative values indicate values smaller in OpenFAST than the nominal/experimental values.

| Mass OpenFAST vs Nominal | | Natural Frequency OpenFAST vs Experimental | | Distance OpenFAST vs 3D Scan | |
|---|---|---|---|---|---|
| Blade | -6 % | First rotor flap | +3 % | Shaft tilt | -22 % |
| Hub mass | -18 % | First rotor edge | -4 % | Overhang | 0 % |
| Rotor mass | -12 % | First tower fore-aft | +9 % | Clearance at blade tip | -11 % |
| Nacelle mass | -14 % | First tower side-side | +3 % | | |
| Tower-top mass | -12 % | | | | |

After the model validation, OpenFAST was coupled to the industrial controller provided by GE and was run for 3196 simu-
lations modeling inflow conditions prescribed by the IEC 61400-1 (2019) standards as well as with extreme inflow conditions
observed at the NREL Flatirons Campus (Hamilton and Debnath, 2019). Table A5 lists all the DLCs run in preparation of the
180 experiment, whereas Figures A1, A2, A3, and A4 show the load rankings for the blade root combined moment, low-speed
shaft combined moment, tower-top combined moment, and tower-base combined moment, respectively. OpenFAST predicts
that the loads at the blade root and tower-base are dominated by storm case DLC-6.2 and do not increase in the downwind



configuration. Combined moments on the drivetrain and at tower-top could increase up to +20 % by converting from upwind to downwind. However, the increases correspond to load cases that are unlikely to occur, and conversations with the manufacturers of the different nacelle components helped the team conclude that the risk of causing a failure in the drivetrain components by exceeding the ultimate loads was sufficiently low, thus allowing the team to safely proceed with the experiment.

### 3.1.3 Reversed aerodynamic thrust

The risks of operating under reversed aerodynamic thrust received separate attention. Reversed thrust generated concerns because although the drivetrain components are designed to resist some amount of reversed thrust even during upwind operations, the design of the components is not symmetric. Attention was paid to the front cover of the main bearing. In upwind operations, thrust at the main bearing is received by a shoulder in the housing and then transmitted to the bedplate. There is no shoulder, however, in the upwind direction. In the event of reversed thrust, whether upwind during a shutdown or because of downwind operations, the full aerodynamic thrust is received by the front cover, which is a metal plate bolted to the main bearing housing. The team conducted a finite element analysis of the main bearing loaded under a reversed thrust of 300 kN. The value was set 50 kN higher than the ultimate thrust of 250 kN. The analysis showed minimal stress, with strain levels concentrated close to the areas in the front cover next to the bolts, but still well below yield limits. The team then decided to install a temperature-compensated strain gauge half-bridge on the front cover of the main bearing. One strain gauge was placed in the radial direction and monitored the radial strain of the cover, whereas the second strain gauge was placed along the circumference and compensated the first gauge for temperature effects. After the installation, the turbine in the upwind configuration was subjected to an emergency shutdown while operating with average wind speeds of 11 m s$^{-1}$, which roughly corresponds to maximum aerodynamic thrust. OpenFAST predicted a peak negative thrust of 50 kN during the maneuver, but minimal strains were recorded by the gauges. After this test, the team proceeded with the commissioning of the downwind experiment and committed to closely monitoring the strains during the downwind operations.

### 3.1.4 Reduced blade–tower clearance

Converting a downwind rotor from upwind to downwind requires pitching the blades by 180°. When blades are prebent like those of the GE 1.5 MW wind turbine, downwind will bring the blade tips closer to the tower as opposed to farther from the tower. The results of the OpenFAST simulations in terms of blade–tower clearance as a function of wind speed are shown in Figure A5, which depicts how the minimum clearance from the upwind rotor was clearly violated by operating the rotor in the downwind configuration. The team addressed this major risk with a set of actions. First, they decided to limit data collection to 10 min average wind speeds at hub heights up to 13 m s$^{-1}$, which helps avoid those conditions where the blades are pitched and unloaded and therefore fly closer to the tower. Next, 11 laser sensors were installed around the tower at a height of 45 m. The sensors measured in real time the distance between the tower and the blade section located 5 m away from the blade tip. Once the sensors were installed, the numerical predictions could be validated.

During the precommissioning of the downwind rotor, the team placed one blade at a pitch of 0° right in front of the tower. The recording of the laser sensor read 3.2 m, which was 18 cm less than what OpenFAST predicted. This mismatch could be





partially explained by the mismatch between the nominal and actual shaft tilt, as discussed in Section 3.1.2. Figure 3 shows the comparison between numerical predictions from DLC-1.1 and DLC-1.3 and the experimental recordings obtained by the lasers during 1 day of upwind operations on 7 Feb. 2024. The comparison returned a satisfactory match. Once the confidence in the model was established, the team ran some worst-case scenarios for the downwind rotor, which generated a clearance

of 0.8 m. The team also plotted the clearance in terms of probability density functions; see Figure A6. From the probability density functions, the risk of a tower strike assuming normal operation and no additional faults, such as controller fault or blade structural failure, was quantified to be equal to 1E-13.

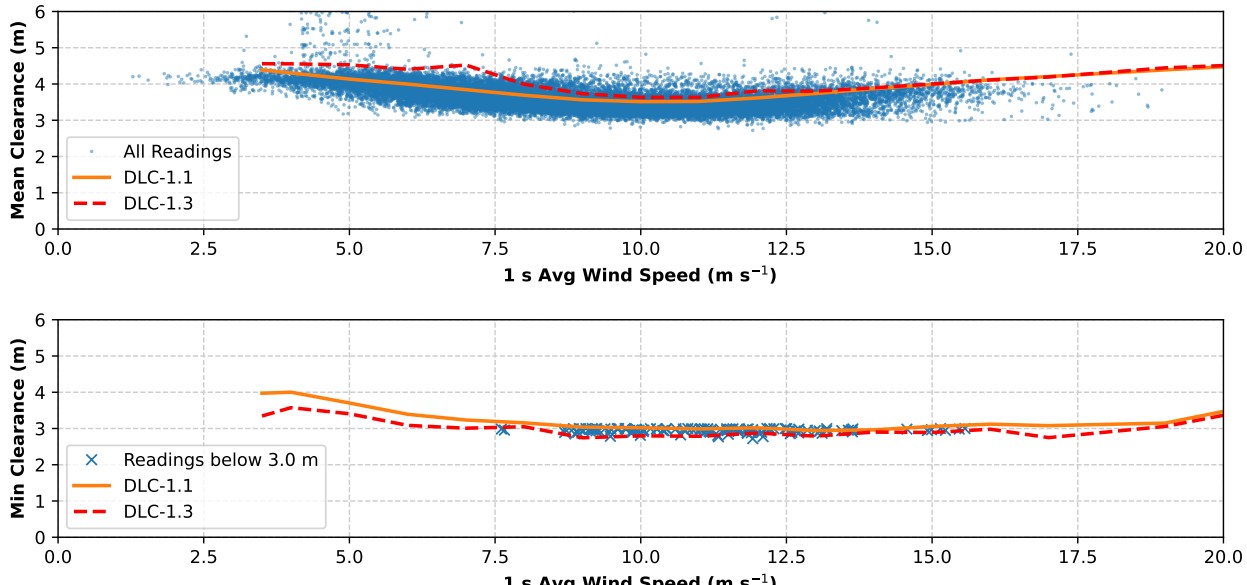

**Figure 3.** Comparison between the numerical mean and minimum blade–tower clearance from six seeds of DLC-1.1 and DLC-1.3 and the experimental readings over 24 hours for the upwind rotor of the GE 1.5 MW wind turbine. The simulations were performed with nominal nacelle tilt.

The last step to lower the risk of a potentially catastrophic tower strike during the downwind experiment consisted of quantifying the occurrence of DLC-1.4, which corresponds to an extreme coherent gust with direction change (ECD) (IEC

61400-1, 2019). The team analyzed 10 years of data collected between 2014 and 2024 at the M5 met mast installed at the NREL Flatirons Campus. No gust was found exceeding the threshold defined by IEC 61400-1 (2019), namely a gust of 15 m s$^{-1}$ combined with a change of direction. The magnitude of the gust is constant across wind speeds, whereas the magnitude of the change of direction depends on the wind speed. Below 4 m s$^{-1}$, the change is set to 180°. Above 4m/s, the prescribed direction change is smaller, as shown in Figure 4, which also shows the experimental data points that were closest to an ECD event. The

team simulated two of these points in OpenFAST by generating a coherent wind with an extreme change in direction, and the





blade–tower clearance was equal to 1.6 m. On the basis of these results, the team concluded that it was safe to proceed with the experiment.

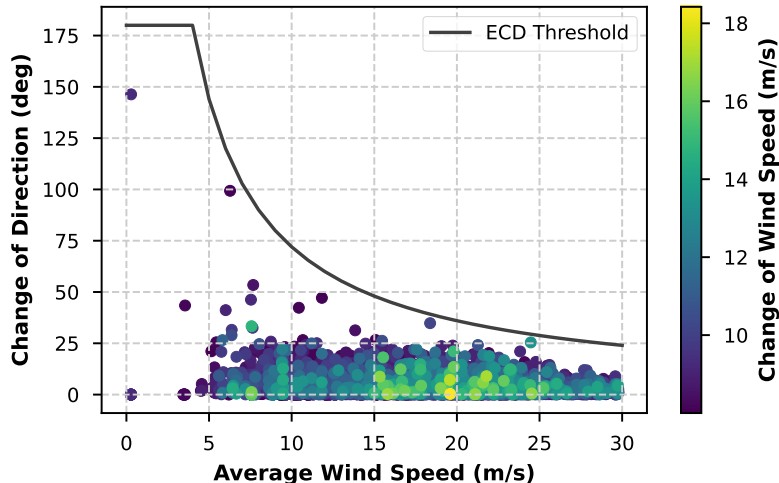

**Figure 4.** The markers indicate ECDs recorded between 2014 and 2024 at the M5 met mast installed at the NREL Flatirons Campus. The black solid line represents the ECD as defined in IEC 61400-1 (2019). The definition of IEC 61400-1 (2019) is found to be conservative, and no gust is found exceeding the threshold.

## 3.2 Conversion to downwind

The team at NREL does not have the ability to change the control algorithm of the GE 1.5 MW wind turbine. However, the
235 experiment was possible thanks to a few changes in the hardware of the machine that made the controller blind to the downwind orientation of the rotor. The first step consisted of applying a 180° offset to the three blades. Each blade is equipped with two bumpers that mark the 0° (rated) and 90° (parked) positions of the blade pitch angle. During the standard calibration process, a limit switch at the blade root passed the two positions to the controller. The team then manufactured six new pitch bumpers that were identical to the six existing pitch bumpers. The six new bumpers were then glued onto the nuts of the blade root
t-bolts at 180° and 270°. The standard pitch calibration procedure was then started at the 180° mark. This procedure allowed the team to achieve the offset in blade pitch, which was later validated with a photogrammetry process relying on photos shot from the ground while pointing vertically up. Table 3 lists the three pitch angles for both upwind and downwind operations, reconstructed via photogrammetry.

The next step consisted of physically turning the wind vane by 180°. The team evaluated the possibility of intercepting and
245 retuning the signal coming from the upwind vane but opted for a mechanical approach to avoid the risk of falling into the deadband of the vane instrument. Although the team is aware that upwind-oriented wind vanes commonly implement an offset





**Table 3.** Results of the photogrammetry carried out to ensure the alignment of the blade pitch among the three blades.

| Pitch (°) | B1 | B2 | B3 |
|---|---|---|---|
| Upwind | 1.32 | 1.04 | 1.29 |
| Downwind | 2.17 | 1.56 | 2.20 |

of a few degrees to compensate for the downwash effects of the upwind rotor (Simley et al., 2021), this offset was neglected during the downwind experiment.

The third step consisted of switching the phases of the generator, which spun backwards in downwind, to enable the correct application of the generator torque. This step was less problematic than the others as the original gearbox of the turbine had the high-speed shaft spinning counterclockwise, and the generator of the upwind rotor spun counterclockwise until the gearbox was replaced in 2018 with a new one whose high-speed shaft spun clockwise, like the low-speed shaft.

The fourth and last step involved installing a supervisory controller. The controller computed the 1 s and 10 s averages of the wind speed measured on top of the nacelle and on the met mast at hub height. The supervisory controller also monitored the clearance between the blades and tower. Throughout the experiment, the research team could set thresholds to one or more of these quantities. When a threshold was violated, a small actuator located at the base of the tower mechanically pushed the idling command, sending the rotor to idling. The idling command was chosen as it triggered the mildest of the shutdown maneuvers, gradually reducing the rotor speed and unloading the three blades.

### 3.3 Commissioning

The experiment was conducted only in attended mode. The first commissioning test consisted of spinning the generator to minimum rpm, which is roughly 1000 rpm, and then passing the safety checks of the GE controller. The second commissioning test consisted of connecting the generator to the grid. This step was performed in a low wind day, when wind speeds were below 5 m s$^{-1}$. After that, the turbine was ready for the actual testing campaign. During the testing, the team monitored:

- blade–tower clearance

- temperature signals from gearbox and generator

- main bearing front cover strain

- particle counter in the gearbox oil.

The team operated the turbine only when winds came consistently from the northwest sector. The team did not operate the turbine when it was subjected to winds generated by local storms, which can move erratically along the plains surrounding the test site.





### 3.4 Experimental data collection

This section describes the data collection for loads (Section 3.4.1) and acoustics (Section 3.4.2).

#### 3.4.1 Mechanical loads

Data collection for mechanical loads followed the guidance prescribed by IEC 61400-13 (2015). The data acquisition system
is based on a National Instruments PXI real-time scan engine paired with a custom-developed and validated LabVIEW-based
software, which synchronously records all samples with GPS time stamps. Data are saved in two parallel files: 24 hour files
using a 1 Hz sample rate and 10 min files using a 50 Hz sample rate. To arrive at a valid dataset, the following filters were
applied to both the upwind and downwind data:

- Data were removed when minimum power production was less than 0 kW.

- Data were removed if the mean wind direction was outside of the measurement sector (243°–310°).

- Data were removed when the mean wind speed was below the cut-in (3.5 m s$^{-1}$).

- Data were removed when the data acquisition system was not functioning normally.

- Data were removed when critical instrumentation used in the experiment was not functioning normally.

Data was collected during the time periods specified in Table 4 for each respective dataset. Figure 5 provides the resulting
capture matrices, showing the number of 10 min time periods categorized by wind speed and turbulence intensity for both the
upwind and downwind datasets.

**Table 4.** Time periods (expressed as year-month-day) and number of 10 min data samples of downwind and upwind data collection.

|          | Start      | End        | 10 Min Samples |
|----------|------------|------------|----------------|
| Downwind | 2024-04-13 | 2024-06-04 | 41             |
| Upwind   | 2024-06-29 | 2024-07-19 | 96             |

#### 3.4.2 Acoustics

The acoustic data acquisition system was a National Instruments cRIO-based system that stored raw sound pressure data at
51.2 kHz. Data were GPS-timestamped to allow synchronization with the turbine data acquisition system. The data acquisition
system and microphone setup was a subset of the setup adopted in Hamilton et al. (2021). Recordings of the calibration tones
were made at the beginning and end of every measurement period and compared to ensure minimal drift in acoustic values
during the course of the experimental campaign.

Although attempts were made to collect acoustic data on multiple days, only the day of 13 April 2024, had favorable winds
and less interrupting noise due to construction being stopped on a Saturday. The analysis will focus on that day as a result.




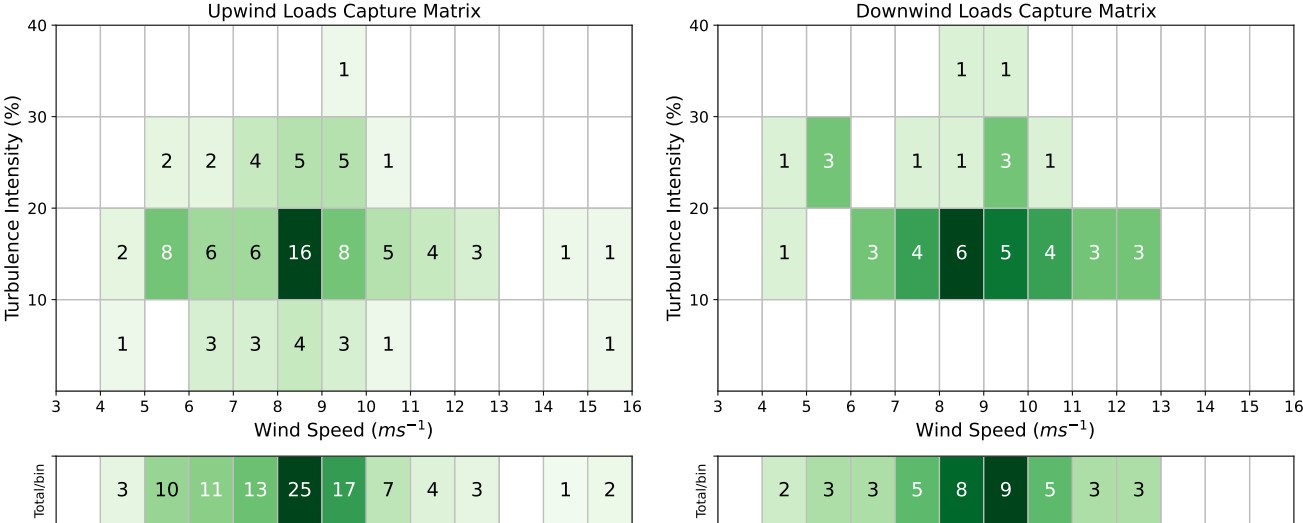

**Figure 5.** Capture matrices for the upwind (left) and downwind (right) datasets.

## 4 Experimental and numerical data analysis

This section describes the data processing of the experimental and numerical datasets for both loads and acoustics.

### 4.1 Loads

#### 4.1.1 Experimental data

The tower loads are calculated in the nacelle reference coordinate system as fore-aft and side-side bending moments using the yaw position. As per IEC 61400-13 (2015), the method of bins was utilized to determine bin averages and bin standard deviations for operating loads and damage equivalent loads (DELs). All data were binned by wind speed with 1 m s⁻¹ widths starting at 3 m s⁻¹ and ending at 16 m s⁻¹. This upper wind speed limit was governed by the downwind operating limitations that were set as described in Section 3.1.4. The DEL was computed in accordance with IEC 61400-13 (2015):

$$DEL = \left( \frac{1}{N} \sum_{i=1}^{n} N_i \cdot F_i^m \right)^{\frac{1}{m}} \tag{1}$$

where $N$ is the total number of cycles, $n$ is the number of load ranges, $N_i$ is the number of cycles at load range $i$, $F_i$ is the amplitude of load range $i$, and $m$ is the Wöhler exponent. The DELs for the operating moments were calculated using material slopes $m$ typical for the component and without Goodman correction. $m$ was set to 4 for the tower and main shaft loads and to 10 for the blades. An exponent of 4 is typical for steel and cast iron, whereas an exponent of 10 is more common for fiberglass





and other similar composites used in blades. A four-point rainflow counting algorithm based on the guidance from Amzallag
et al. (1994) was used to analyze the high-frequency time series data for calculation of the short-term DELs.

### 4.1.2 Numerical data

The two OpenFAST models of the upwind and downwind configurations used during the FMEA and described in Section 3.1.2
were used for the generation of the numerical predictions for power and loads. OpenFAST was run with two different sets of
inflows. First, for every 10 min experimental data point, six turbulent seeds were run in OpenFAST. Each set of six inflows
was generated by running the solver TurbSim (Jonkman, 2014) matching the average wind speed, average turbulence intensity,
and average exponential shear exponent. OpenFAST was then run with the air density corresponding to each data point. This
approach follows that described in Brown et al. (2024) and resulted in 246 OpenFAST simulations in the downwind case and
612 simulations in the upwind case. Note that this modeling approach aims to minimize the differences between numerical
predictions and experimental observations, although it also inherently leads to numerical results where upwind and downwind
face different inflow conditions, and the difference in loads and performance discussed later in Section 5.1 inevitably blend
differences generated by the two orientations of the rotor with differences coming from the different inflow conditions. This
approach however helps to focus on the validation of the numerical predictions. To isolate the effects of the rotor orientation, a
second set of simulations was then run modeling six turbulent seeds of the normal turbulence model prescribed by IEC 61400-1
(2019) in DLC-1.1 between cut-in and cut-out wind speeds in steps of 2 m s$^{-1}$, for a total of 72 simulations per rotor orientation.
In this second set of simulations, upwind and downwind rotors face the same exact inflow. The approach followed in this second
set of simulations is the same as the one often followed in existing literature, such as Bortolotti et al. (2019, 2022).

### 4.2 Acoustics

The analysis of acoustic was restricted to the experimental data, which were all listened to and quality controlled. Anything not
directly part of the normal ambient acoustic signature of the turbine or background was disqualified from analysis. Interruptions
that were removed included noise from wildlife (such as birds, frogs, and grasshoppers) and noise from human activities (such
as aircrafts and helicopters flying near the test site, cars or motorcycles riding along the nearby roads, and trains riding along
the railway). Figure 6 includes a histogram of valid data collected as a function of the wind speed bin. The red line represents
the minimum required data in each bin according to the standard (IEC 61400-11, 2018). At certain wind speeds, the number of
data points in the aeroacoustic dataset did not meet the minimum threshold required by the standards. This limitation is well
known in the study. Nevertheless, we proceeded with the analysis.

During the experiment, an intermittent but audible amplitude modulation was observed, and data were processed to quantify
metrics describing it. Amplitude modulation analysis followed the method of IEC 61400-11-2 (2024) and Bass et al. (2016). It
incorporated the following primary steps:

– Calculate one-third-octave slices for the data in question with a period of 100 ms

– Sum the bands of interest (50–200 Hz, 100–400 Hz, 200–800 Hz)



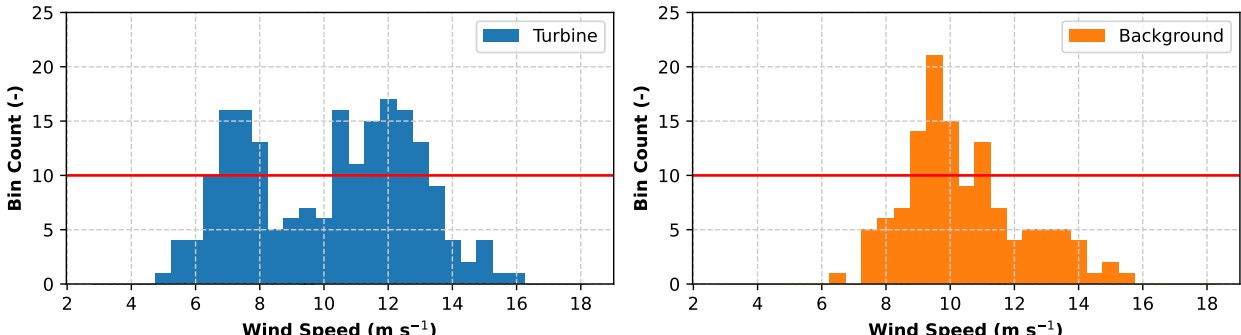

**Figure 6.** Histograms of valid turbine and background data.

– Analyze the band of interest in amplitude modulation identified by the model provided by the AMWG (UK Institute of Acoustics Amplitude Modulation Working Group, 2017).

Additional bands below 50–200 Hz were investigated, but it was found that the amplitude modulation energy was most prevalent in the 50–200 Hz band.

The model provided by the AMWG is documented in IEC 61400-11-2 (2024). When amplitude modulation was found to be present by the model, the output included the fundamental frequency in hertz, the prominence ratio, and the modulation depth in decibels. The number of occurrences as a function of the number of 10 s averaged samples could be calculated by comparing the number of times the model identified amplitude modulation to the number of samples presented to the model. The fundamental frequency was the dominant frequency of amplitude modulation found by the amplitude modulation detection method. Prominence ratio $p_{AM}$ is defined as the level to which the amplitude modulation stands out compared to the surrounding levels. The average prominence ratio was computed as defined by IEC 61400-11-2 (2024) and is shown in Eq. (2).

$$p_{AM} = \frac{L_{pk}}{L_m} \qquad (2)$$

where $L_{pk}$ is the magnitude of the fundamental peak and $L_m$ is the masking level. Both $L_{pk}$ and $L_m$ are expressed in decibels; therefore, $p_{AM}$ is nondimensional. Lastly, the modulation depth is defined as the distance between the peak and valley of the modulation of the sound pressure level.

## 5 Results

This section describes the results for loads (Section 5.1) and acoustics (Section 5.2).





## 5.1 Loads

The experimental data and the numerical power and load data can be compared in a number of ways. In this paper, we organized
a first comparison in terms of two-by-two plots. The top-left quadrant shows a scatterplot of the raw experimental and numerical
data for both the upwind and downwind configurations. The top-right corner shows the result of data binning. The bottom-left
plot shows the difference in binned data between the downwind and upwind cases (downwind minus upwind), where the first
line compares numerical data and the second line compares experimental data. The bottom-right plot switches the comparison
and shows the difference in binned data between experimental and numerical data. The first line compares upwind data, and
the second line, downwind data. Note that scatterplots were intentionally chosen over uncertainty bands for the binned data
to maximize the clarity of the conclusions. Note also that the OpenFAST data refer to the simulations that model the inflow
recorded in the field. The bottom-left plot show however also the comparison between numerical predictions for upwind and
downwind under DLC-1.1, as discussed in Section 4.1.2.

Figure 7 shows the active power of the generator. The bottom-left plot shows that both OpenFAST predictions and experimental recordings oscillate around the 0 % line, but the solid light-green numerical line is consistently lower than the dashed
dark-green experimental line, which sits above 0 % except at 7.5 m s$^{-1}$. In other words, the experimental recordings show
net gains for the downwind rotor compared to the upwind rotor. These gains are not clearly visible numerically and can be
compared to the dash-dotted black line that shows that, during DLC-1.1, OpenFAST predicts a small drop in power for the
downwind rotor. This prediction is consistent with literature, see for example Bortolotti et al. (2019, 2022) and references
therein. Consistently with the bottom-left plot, the bottom-right plot shows that for both the upwind and downwind configurations, experimental recordings are higher than the OpenFAST predictions, with the downwind case returning a larger error.

Figure 8 shows the DEL of the blade root flapwise moment. Statistics are averaged across the three blades. The top-left
plot shows a notable spread in this quantity, which is impacted by the turbulence intensity. The bottom-right corner shows
that OpenFAST is underpredicting this quantity by as much as 100 % where few data points are available and between 10 %
and 30 % in the bulk of the dataset, namely between 6.5 and 12.5 m s$^{-1}$. Still, the top-right and bottom-left plots show that
numerical predictions and experimental observations consistently return an increase in the DEL between 10 % and 20 % for
the blade root flapwise moment in the wind speed range of 6.5 and 12.5 m s$^{-1}$. The prediction of increasing DEL matches with
both the inflow from the field and the inflow from DLC-1.1, which is represented by the dash-dotted black line.

Figure 9 shows the DEL of the blade root edgewise moment. The takeaways are qualitatively similar to the ones for the DEL
of the blade root flapwise moment, with a few key differences. The top-left quadrant shows a smaller spread in the data, and
the bottom-left plot shows an increase in the DEL for the downwind rotor between 2 % and 10 % depending on the wind speed.
Still, numerical predictions and experimental observations match in terms of trends. Also, experimental recordings are again
higher than the predictions from OpenFAST.

Last, Figure 10 shows the DEL of the tower-base fore-aft moment. The takeaways are again similar to the ones from Figure 8.
The scatterplot in the top left shows a high variability of the data caused by turbulence. The binned averages reported in the
top-right and bottom-left plots show an increase in the DEL between 10 % and 20 % between 6.5 and 12.5 m s$^{-1}$. The increase



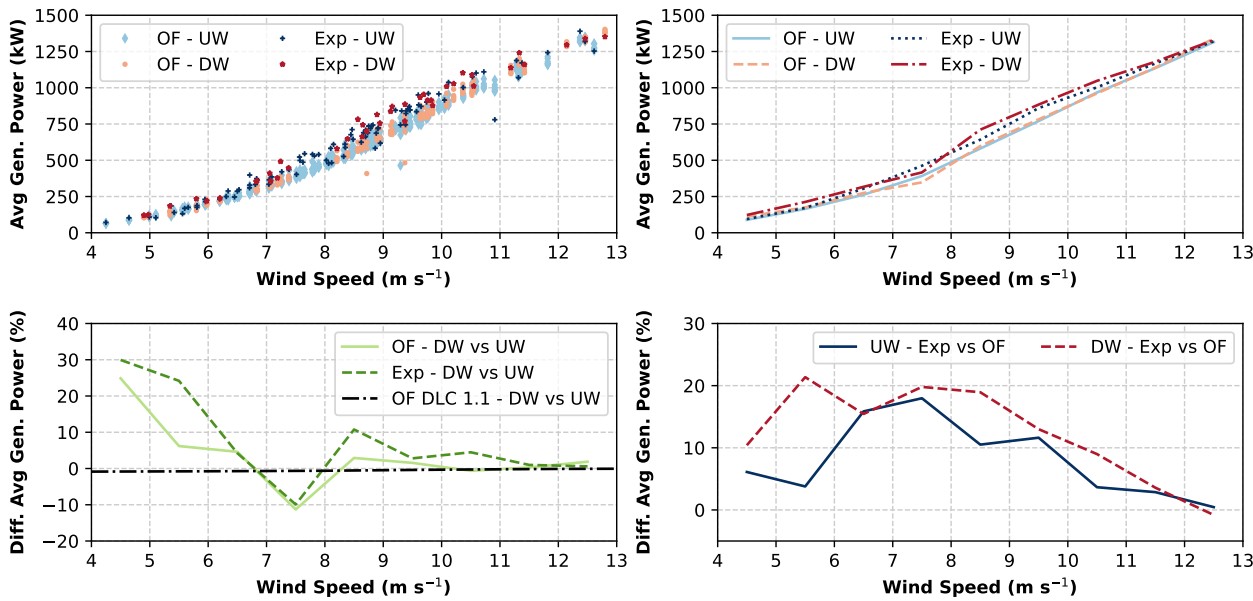

**Figure 7.** Power of the upwind (UW) and downwind (DW) configurations in OpenFAST (OF) and in the field (Exp).

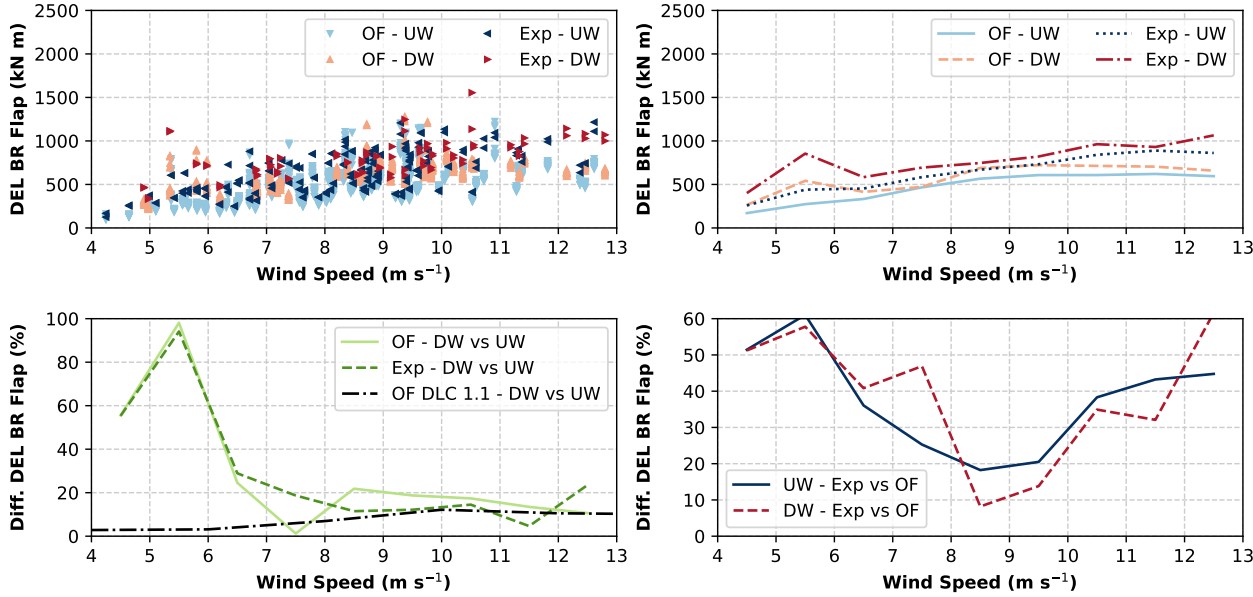

**Figure 8.** DEL of the blade root (BR) flapwise moment of the upwind (UW) and downwind (DW) configurations in OpenFAST (OF) and in the field (Exp).



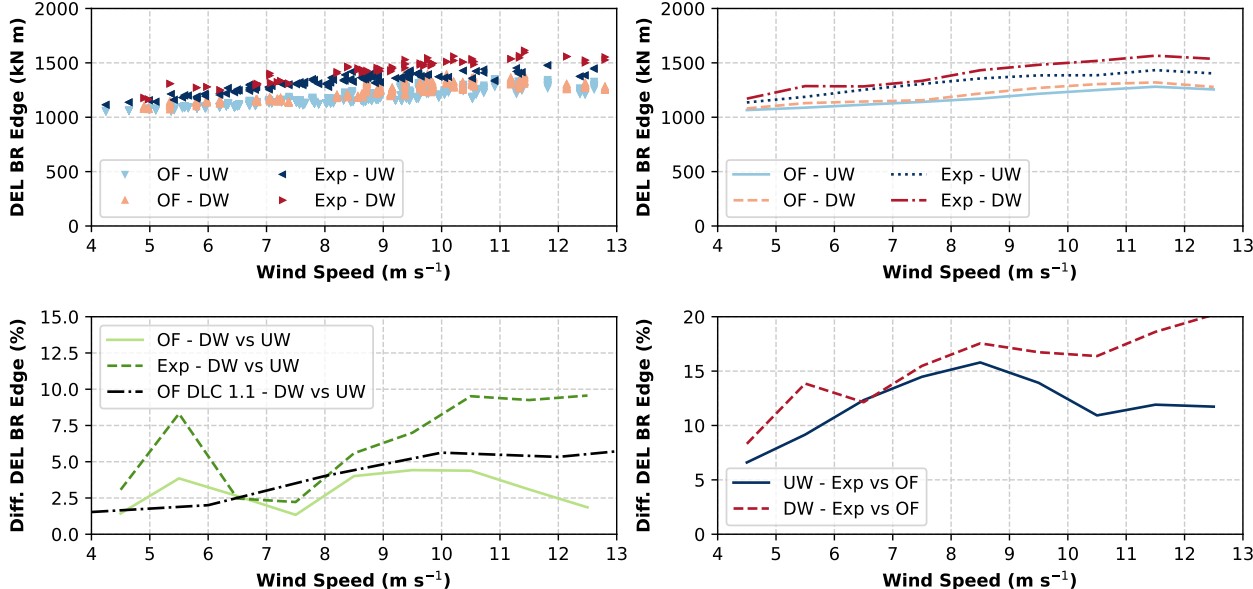

**Figure 9.** DEL of the blade root (BR) edgewise moment of the upwind (UW) and downwind (DW) configurations in OpenFAST (OF) and in the field (Exp).

in the solid light-green line is higher than the prediction of OpenFAST using DLC-1.1 inflow conditions and suggest that results are influenced by turbulence. However, the trend of increasing DEL for downwind is confirmed.

These figures represent just a subset of all possible visualizations of the data. More plots are provided in the appendix.
Figures A7 and A8 show the comparisons for the average blade root flapwise and edgewise moments, respectively. Figures A9 and A10 show the comparisons for the average tower-base fore-aft and side-side moments, respectively, whereas Figure A11 shows the comparison for the DEL of the tower-base side-side moment. These plots are less conclusive than the ones presented in the main body of this article but are still included for completeness.

An additional comparison is provided in Table 5, where power and DELs of blade root flapwise, blade root edgewise, and
400 tower-base fore-aft moments are weighted by a Weibull probability density function and integrated between wind speeds of 4.5 and 12.5 m s⁻¹, which is the range of wind speeds with the highest density of data points. The Weibull probability density function is modeled with a shape factor of 2 and a mean wind speed of 7.5 m s⁻¹, which corresponds to a scale factor of 8.46 m s⁻¹. The first column of the table shows the comparison between upwind and downwind for the numerical predictions, whereas the second column shows the comparison for the experimental recordings. The trends discussed in Figures 7, 8, 9, 10
are consistently condensed into a single number. OpenFAST predicts that the downwind rotor operating in the field generates 0.5 % more power, whereas the experimental recordings return a higher gain of +3.8 %. For the DEL of blade root flapwise moment, OpenFAST predicts a growth of 24.7 %, and the data from the field shows +25.7 %. For the DEL of blade root edgewise moment, the growth in OpenFAST is +3.1 %, and the field shows +5.9 %. Lastly, the DEL of tower-base fore-aft





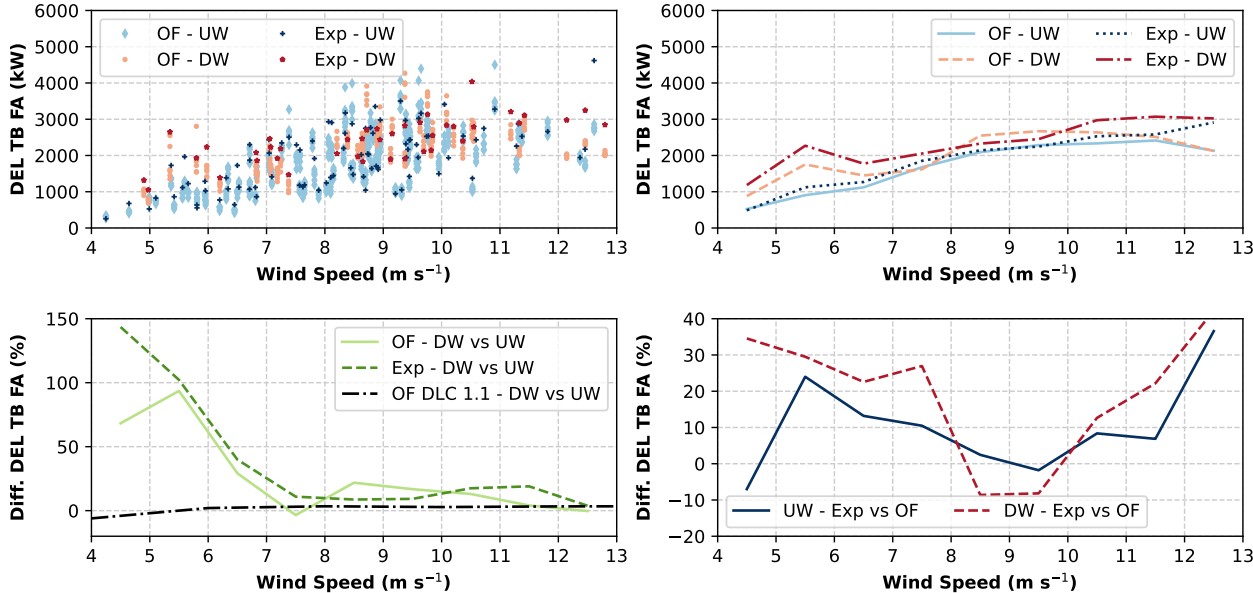

**Figure 10.** DEL of the tower-base (TB) fore-aft (FA) moment of the upwind (UW) and downwind (DW) configurations in OpenFAST (OF) and in the field (Exp).

moment grows by 21.1 % in OpenFAST and by 26.7 % in the field. The third column in Table 5 shows the comparison

downwind vs upwind for the predictions of OpenFAST under the inflow of DLC-1.1. In this case, the Weibull probability density function extends from cut-in wind speed to cut-out wind speed. The changes in power and DELs, which are consistent with literature, isolate the effects of the rotor orientation as predicted by OpenFAST and eliminate the effects of different inflow conditions.

**Table 5.** Comparison between downwind and upwind of generator power and key DELs weighted by Weibull probability density functions.

| Metric | OF | Exp | OF - DLC 1.1 |
|---|---|---|---|
| Avg Gen. Power | +0.5 % | +3.8 % | −0.4 % |
| DEL BR Flap | +24.7 % | +25.7 % | +7.9 % |
| DEL BR Edge | +3.1 % | + 5.9 % | +4.0 % |
| DEL TB FA | +21.1 % | +26.7 % | +2.5 % |

These results indicate that OpenFAST is capable of predicting the loads of both upwind and downwind rotor of the GE

1.5 MW wind turbine fairly accurately, especially in terms of trends. The experimental measurements confirm the prediction of higher DELs for downwind, although suggest that the numerical predictions might be slightly underestimating the difference





between upwind and downwind rotors. For power, OpenFAST predicts a small influence of the downwind orientation of the rotor, whereas the experimental measurements show a power increase.

## 5.2 Experimental acoustics

The acoustics dataset was processed first in terms of overall sound pressure and sound power levels, as discussed in Section 5.2.1, and then in terms of amplitude modulation, as discussed in Section 5.2.2.

### 5.2.1 IEC results

Valid turbine and background raw sound pressure levels plotted as a function of hub-height wind speed are shown in the top plot of Figure 11. The downwind dataset is compared to the upwind dataset collected during an IEC noise test conducted in 2011

and documented in Roadman and Huskey (2015). Data largely overlap. Following the analysis method defined in IEC 61400-11 (2018), the bottom plot in Figure 11 shows the averaged overall sound power levels as a function of wind speed at a height of 10 m. The data show strong agreement in all but the highest wind speed bin between upwind and downwind measurements, although the difference at higher wind speeds might be simply caused by sparsity of data points. After this comparison was completed, the team switched focus to compare the amplitude modulation to distinguish any acoustical differences between

the upwind and downwind configurations.

### 5.2.2 Amplitude modulation

As amplitude modulation represented the characteristic acoustic behavior of the turbine during operation downwind, the analytical focus of the acoustic investigation centered around amplitude modulation.

First, the data were plotted in the form of spectrograms, which offer a qualitative visualization of the dataset. Figure 12

shows spectrograms of two 10 s long audio snippets of upwind operational data from 2011 and downwind operational data from the data collected on 13 April 2024. Both snippets were recorded by the microphones that measured in audible range. There is little to no amplitude modulation audible in the upwind data, whereas amplitude modulation is clearly audible in the downwind clip. The spectrograms were generated in Python, leveraging the open-source SciPy library with no filtering nor windowing applied to the data. Although the two spectrograms are similar, the downwind one shows vertical periodic spikes

and striations at an interval corresponding to the blade passing. Vertical striation is less notable in the upwind spectrogram. Note that neither spectrogram shows horizontal striations at low frequency corresponding to the rotor harmonics, such as the spectrograms shown in Blumendeller et al. (2020). Note also that several attempts were made to plot and visualize data with finer discretization along the y-axis as well as in terms of power spectral densities and fast Fourier transforms, but the differences between the upwind and downwind datasets could not be isolated and quantified rigorously.



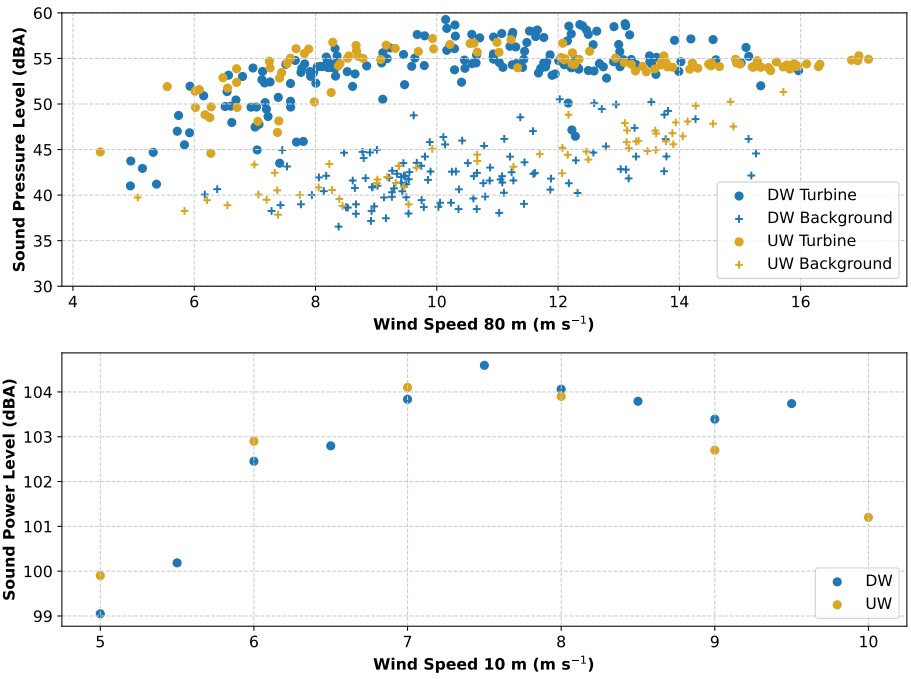

**Figure 11.** (Top) Comparison between sound pressure levels for the upwind (UW) and downwind (DW) rotors. Background data are also included. (Bottom) Comparison in the form of sound power levels.

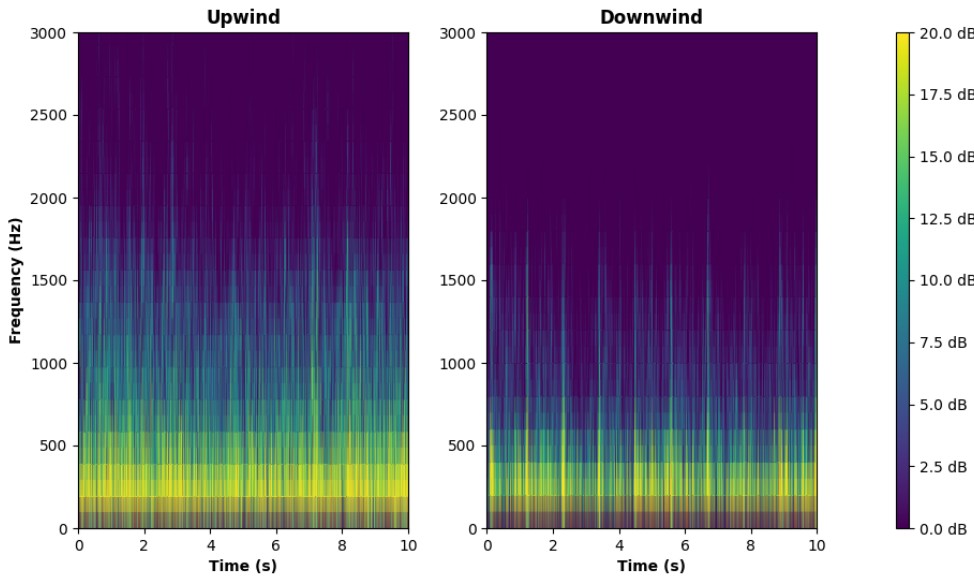

**Figure 12.** Spectrograms of upwind and downwind turbine one-third-octave-band sound pressure levels during normal operation.





Next, data were postprocessed according to the methods for amplitude modulation described in IEC 61400-11-2 (2024), which offer a more quantitative approach than spectrograms. Figure 13 shows a plot of the fundamental frequency as identified from the amplitude modulation method and rotor speed of the turbine as a function of time. When the turbine is operating at a rated speed of 18.3 rpm, the method consistently shows the existence of amplitude modulation and identifies the corresponding fundamental frequency of 0.9 Hz, in line with three times the blade-passing frequency. In contrast, there is no correlation

between the fundamental frequency of the amplitude modulation detected by the model provided by the AMWG and the rotor speed when the 3P frequency of the rotor is below 0.4 Hz, which corresponds to a rotor speed below the minimum operational rotor speed of the turbine.

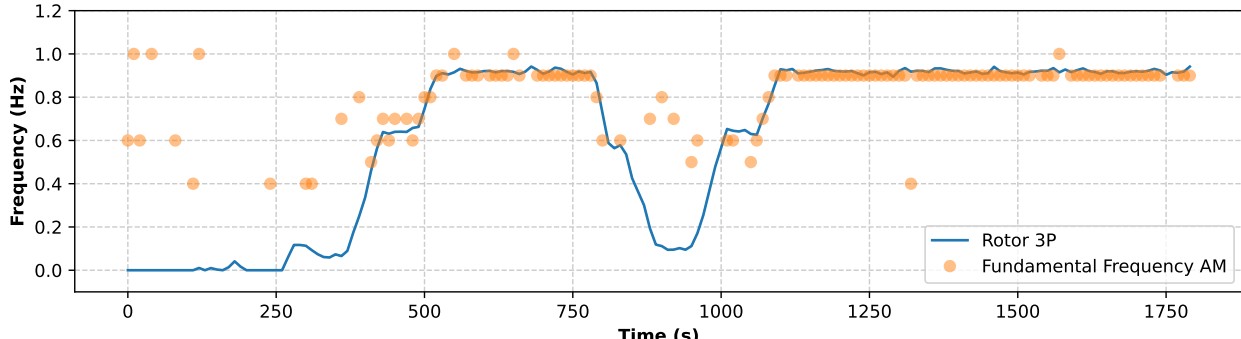

**Figure 13.** Time series of the rotor 3P harmonic, which is the frequency in hertz corresponding to three times the rotor speed, and the fundamental frequency detected by the amplitude modulation (AM) method.

     Three operational cases were studied with the amplitude modulation method: downwind, upwind, and background noise. The background noise was included as a baseline to investigate how often the method identifies false positives. The results are

reported in Figure 14. The data postprocessing identifies amplitude modulation 82 % of the time for the downwind recordings, 29 % of the time for the upwind recordings, and only 7 % of the time for the background analysis. The prominence ratio is 13.5 for the downwind configuration, 2.9 for the upwind configuration, and 1.3 for the background analysis. The modulation depth is found to have a mean of 7.6 for the downwind case, 3.8 for the upwind case, and 3.7 for the background case. These results show that the downwind configuration causes more frequent and stronger events of amplitude modulation.




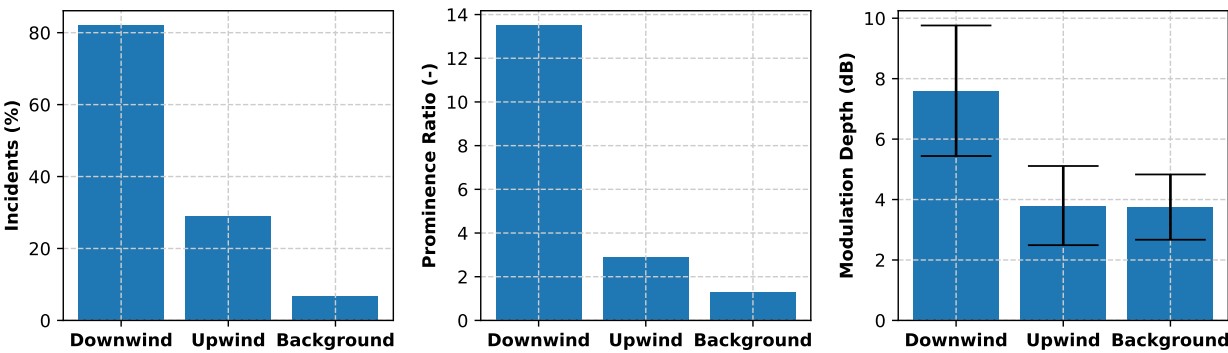

**Figure 14.** Incidents of amplitude modulation (left), prominence of amplitude modulation (center), and maximum, minimum, and mean modulation depths as a function of the operating condition (right).

## 6 Conclusions

This paper discusses the motivation, preparation, risk mitigation, execution, and results of a full-scale experiment where the rotor of a GE 1.5 MW wind turbine was operated in a downwind configuration. In this experiment, the team collected loads, performance, and acoustic measurements. Power and loads are compared to numerical predictions from the aeroelastic solver OpenFAST, which is run both modeling the inflow conditions measured in the field and the standard inflow conditions corresponding to the normal turbulence model prescribed for DLC-1.1 by IEC 61400-1 (2019). The key takeaways are summarized here:

– Given the inflow measured in the field, OpenFAST predicts an increase of +0.5 % in the Weibull-weighted power for the downwind rotor. The experimental recordings show a more marked +3.8 %. Under DLC-1.1 inflow, OpenFAST predicts a drop of the Weibull-weighted power of +0.4 % for the downwind rotor.

– The Weibull-weighted DEL of the blade root flapwise moment grows for the downwind rotor by +24.7 % in OpenFAST and by +25.7 % in the field. Under DLC-1.1 inflow, OpenFAST predicts a growth of the Weibull-weighted DEL of +7.9 %.

– The Weibull-weighted DEL of the blade root edgewise moment grows for the downwind rotor by +3.1 % in OpenFAST and by +5.9 % in the field. During DLC-1.1, the growth is +4.0 %.

– The Weibull-weighted DEL of the tower-base fore-aft moment grows for the downwind rotor by +21.1 % in OpenFAST, by +26.7 % in the field, and by +2.5 % during DLC-1.1.

– Experimental overall sound pressure levels are comparable between the upwind and downwind cases, but the downwind scenario shows higher incidents, prominence ratio, and modulation depth of amplitude modulation.





These results build confidence in the ability of OpenFAST to predict the behavior of both upwind and downwind rotors,
although the growth of DEL in downwind rotors is generally slightly underpredicted, whereas OpenFAST seems to miss a
positive gain in power for the downwind rotor.

Research is currently ongoing investigating the potential of downwind rotors for offshore applications, especially for floating
wind turbines, where the rotor tilt of downwind rotors compensates the average pitch angle of the floating platform and could
lead to a sizable increase in power performance. Full-scale wind turbine concepts with downwind rotors are already at the
485 prototype stage, and techno-economic analyses aim to shed more light on advantages and drawbacks of downwind rotors for
floating wind applications.

*Code and data availability.* OpenFAST is publicly available at https://github.com/OpenFAST/openfast, but the input files modeling the GE
1.5 MW wind turbine are currently not available in the public domain. The NREL team is able to share the experimental datasets that were
collected during the downwind experiment, both the turbine quantities from the SCADA and the recordings from the acoustic array in the
490 field. If interested, please contact the corresponding author.

*Video supplement.* The team collected a number of photos and videos during the experiment. The media material is available upon reasonable
request.





**Table A1.** Summary of turbine instrumentation: nonload channels.

| Instrument | Manufacturer | Model Number |
|---|---|---|
| Primary Wind Speed (80 m) | Thies Clima | First Class Advanced |
| Wind Speed 87 m | MetOne | SS 201 |
| Wind Direction | MetOne | SD 201 |
| Air Pressure | Vaisala | PTB101B |
| Air Temperature | MetOne | T200 |
| Availability | | |
| Pitch Angle Blade 1 | | |
| Pitch Angle Blade 2 | | |
| Pitch Angle Blade 3 | | |
| Main Bearing Temperature | | |
| Gearbox High-Speed Bearing Temperature | | |
| Gearbox Oil Sump Temperature | | |
| Generator Bearing Temperature | | |
| Generator Speed | SCADA | N/A |
| Generator Torque | | |
| Yaw Position | | |
| State Fault | | |
| Turbine Power | | |
| Nacelle Wind Speed | | |
| Tower-Top Lateral Acceleration | | |
| Tower-Top Normal Acceleration | | |



**Table A2.** Summary of turbine instrumentation: load channels.

| Instrument | Manufacturer | Model Number |
|---|---|---|
| Tower-Base Bending Fore-Aft | | LWK-06-W250D-350 |
| Tower-Base Bending Side-Side | | LWK-06-W250D-350 |
| Tower-Top Torque | | CEA-06-125UW-350 |
| Tower-Top Bending Fore-Aft | | LWK-06-W250D-350 |
| Tower-Top Bending Side-Side | | LWK-06-W250D-350 |
| Blade 1 Flap Bending | | WK-09-250MQ-10C/w |
| Blade 2 Lead-Lag Bending | | WK-09-250MQ-10C/w |
| Blade 2 Flap Bending | | WK-09-250MQ-10C/w |
| Blade 3 Lead-Lag Bending | Vishay | WK-09-250MQ-10C/w |
| Blade 3 Flap Bending | | WK-09-250MQ-10C/w |
| Blade 1 Lead-Lag Bending | | WK-09-250MQ-10C/w |
| Main Shaft Bending 0° | | LWK-06-250D-350 |
| Main Shaft Bending 90° | | LWK-06-250D-350 |
| Main Shaft Torque | | LEA-06-W125F-350/3R |
| Blade Tower Clearance | SICK | DT50 |

**Table A3.** Summary of instrumentation: aeroacoustics channels.

| Instrument | Manufacturer | Model Number |
|---|---|---|
| Signal Analyzer | Delta Acoustics | noiseLAB Professional or noiseLAB Wind |
| Microphone | | 4964 |
| Preamplifier | Bruel & Kjaer | 2669-L |
| Calibrator | | 4230 |
| Digital Recorder | National Instruments | 9234, custom software |
| Anemometer (10 m tower) | Thies | First Class |



**Table A4.** Summary of FMEA conducted between NREL and third-party consultant Gulf Wind Technology. Only risks characterized by an RPN of 10 and above are reported.

| Description | RPN | Mitigation |
|---|---|---|
| Component failure due to increased ultimate loads, reversed aerodynamic thrust, and opposite loading from atmospheric shear | 25 | The load analysis performed in OpenFAST showed a 2 % increase in maximum tower-top combined moment and a 12 % increase in maximum shaft combined moment. Other components were dominated by storm loads and did not experience an increase. |
| Failure of the main bearing front cover or associated bolts, which are in the downwind thrust load path (a shoulder in the housing receives these loads in the upwind configuration) | 25 | Finite element modeling of the housing under reversed thrust showed stresses within limits. Strain gauges installed on the front cover monitored deformations during the experiment. A supervisory controller shuts down the turbine in case of excessive tower vibration. |
| Risk of tower strike due to blades being prebent toward the tower | 20 | Simulations in OpenFAST returned a minimum clearance of 1.5 m during an ECD. No such condition was ever recorded at the NREL Flatirons Campus. Risk of tower strike was quantified to be 1E-13. |
| Overheating of the generator due to reversed air scoop | 16 | Temperature signals monitored during the experiment. Supervisory controller shuts down the turbine in case of excessive temperatures. |
| Mismatch between numerical models and real turbine | 16 | A validation of the model was conducted in terms of masses, natural frequencies, and performance of the controller. |
| Increase in fatigue loading due to tower shadow effects | 12 | OpenFAST predicted a minor increase in fatigue loading. The increase is not concerning given the few hours of testing. |
| Errors in the watchdog controller | 12 | Watchdog wrapped the existing supervisory controller of the turbine, which was not modified in any of its elements. |
| Failure of the tower due to alignment of the aerodynamic and gravitational moments | 10 | Tower design was driven by storm loads, which are not expected to change. |
| Rotor aerodynamic imbalance due to pitch misalignment in the downwind configuration | 10 | Photogrammetry ensured that blades were aligned within 1° from each other. Impact was on performance rather than on loads. |
| Damage to the gearbox due to reversed rotation and thrust; although the gear teeth are symmetric, the lubrication flows are not | 10 | Gearbox was operated in reverse during factory acceptance tests. Supervisory controller shuts down the turbine in case of excessive gearbox temperature. The oil particle counter was monitored, and regular visual inspections of gearbox parallel stages showed no damage. |





**Table A5.** Summary of design load cases, following the IEC 61400-1 (2019) standards. *1-year and 50-year storm turbulent wind speeds were tuned for the NREL Flatirons Campus (FC) following Hamilton and Debnath (2019). OpenFAST could not complete a few cases, which are listed in parentheses. Those cases were excluded from the load analysis.

| DLC | Wind Speeds (m s$^{-1}$) | Shear (-) | Seeds | # Cases |
|---|---|---|---|---|
| 1.1 Normal operation | 3.5–25 | 0.2 | 6 seeds | 114 |
| 1.1T NREL FC turbulence | 3.5–25 | 0.2 | 6 seeds | 114 |
| 1.1TL NREL FC turbulence, low shear | 3.5–25 | -0.1 | 6 seeds | 114 (2) |
| 1.1TH NREL FC turbulence, high shear | 3.5–25 | 0.6 | 6 seeds | 114 (1) |
| 1.3 Extreme turbulence | 3.5–25 | 0.2 | 6 seeds | 114 |
| 1.3E Emergency shutdown | 3.5–25 | 0.2 | 6 seeds | 114 |
| 1.3S Normal shutdown | 3.5–25 | 0.2 | 6 seeds | 114 |
| 1.4 Coherent gust | 6–16 | 0.2 | 2 directions | 22 |
| 1.5 Power production | 3.5-25 | 0.2 | 2 directions, 2 shears | 76 |
| 4.2 Normal shutdown | 6-25 | 0.2 | 6 seeds, 4 azimuths | 288 |
| 5.1 Emergency shutdown | 6–25 | 0.2 | 6 seeds, 4 azimuths | 288 |
| 6.1 Idling 8° yaw | 42.5* | 0.2 | 6 seeds, 2 azimuths | 12 |
| 6.2 Idling loss of power | 42.5* | 0.2 | 6 seeds, 15 yaw angles | 90 (16) |
| 6.3 Idling 20° yaw | 34* | 0.2 | 6 seeds, 2 yaw angles | 12 (6) |
| 7.1 Idling pitch stuck | 34* | 0.2 | 6 seeds, 2 yaw angles | 12 |





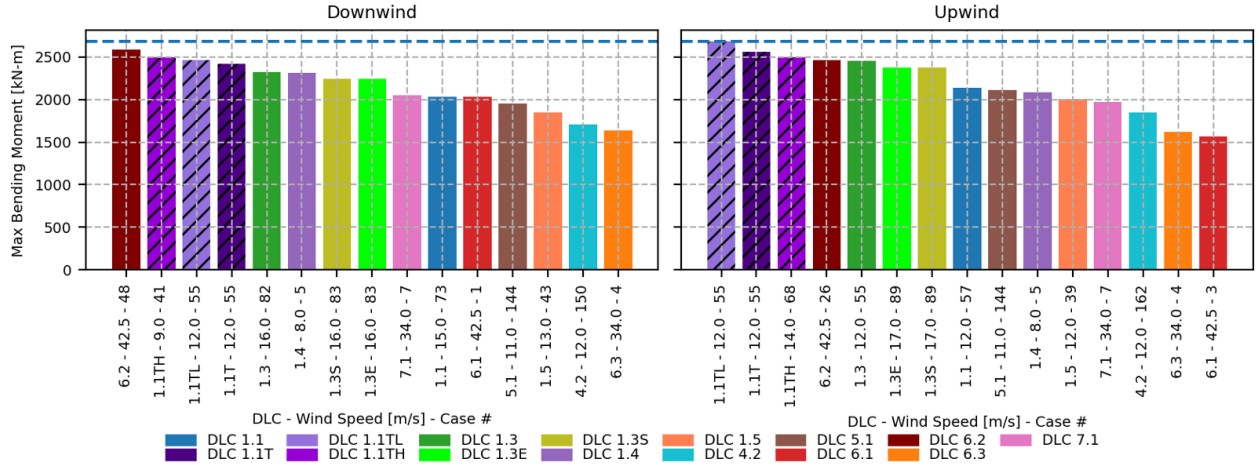

**Figure A1.** Load ranking of the blade root combined bending moment between the downwind (left) and upwind (right) configurations. Downwind operations do not cause a clear increase in this ultimate load, which is dominated by storm case DLC-6.2.

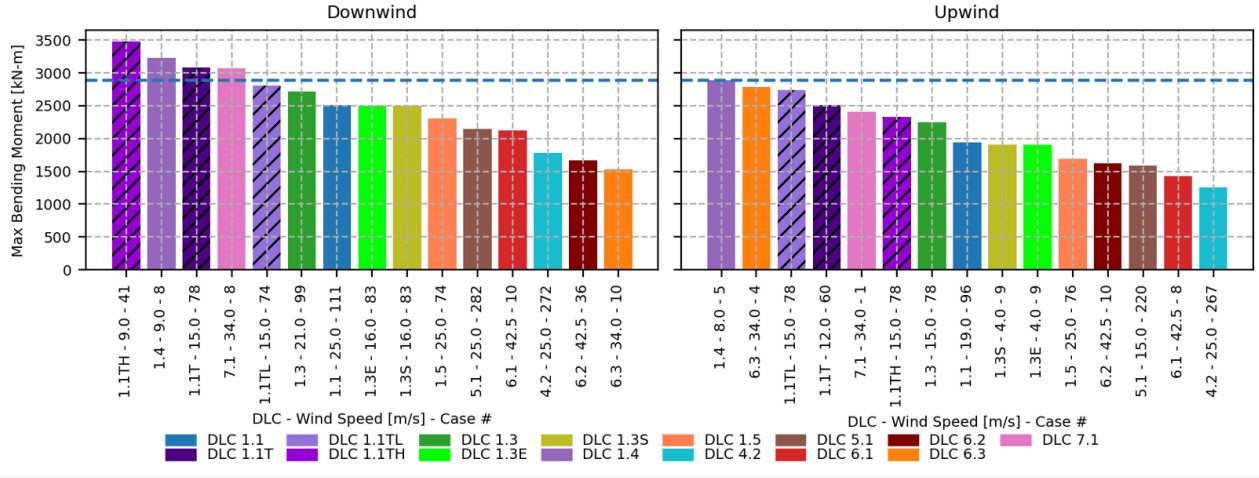

**Figure A2.** Load ranking of the low-speed shaft combined bending moment between the downwind (left) and upwind (right) configurations. Downwind operations cause a +20 % increase in the presence of an unlikely event that combines high turbulence and high shear. If this case is excluded, the increase drops to +12 % and occurs during an unlikely DLC-1.4, namely, an ECD (IEC 61400-1, 2019).

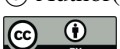

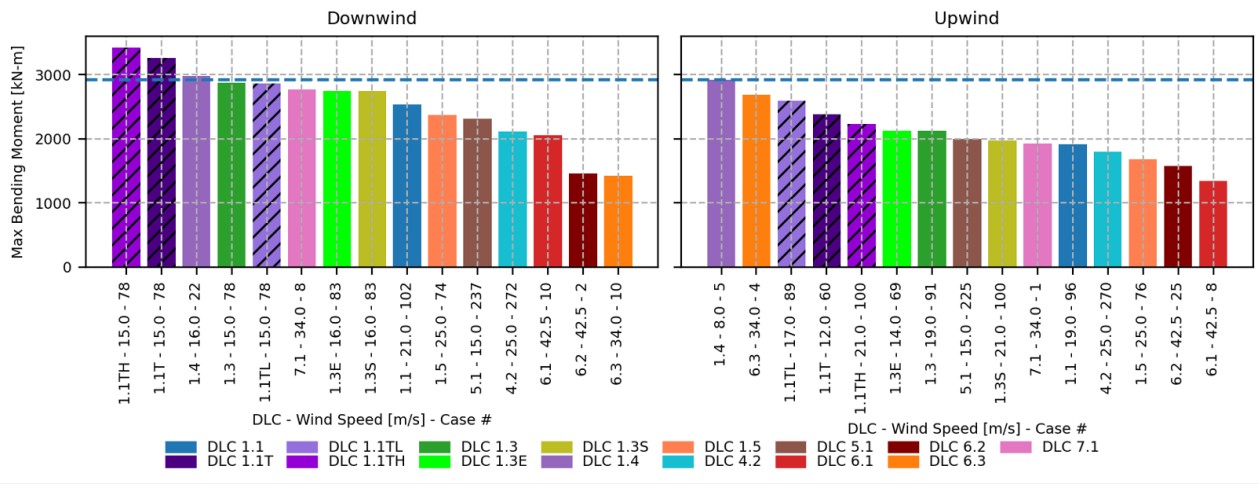

**Figure A3.** Load ranking of the tower-top combined bending moment between the downwind (left) and upwind (right) configurations. Downwind operations cause a +16 % increase in the presence of an unlikely event that combines high turbulence and high shear. If this case is excluded, the increase drops to +2 % and occurs during an unlikely DLC-1.4, namely, an ECD (IEC 61400-1, 2019).

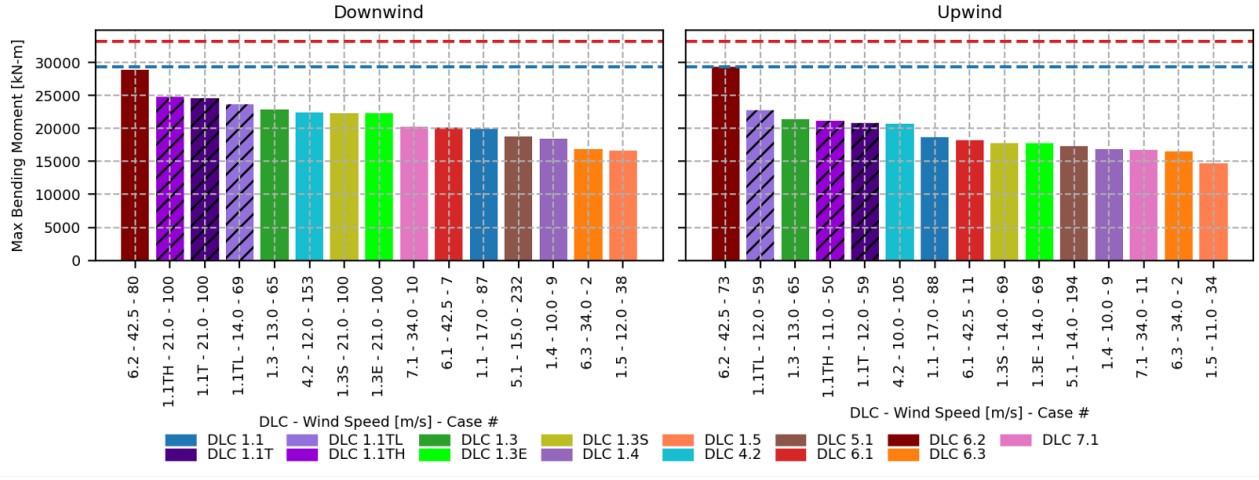

**Figure A4.** Load ranking of the tower-base combined bending moment between the downwind (left) and upwind (right) configurations. Downwind operations do not cause a clear increase in this ultimate load, which is dominated by storm case DLC-6.2. The red dashed line shows nominal design limits.

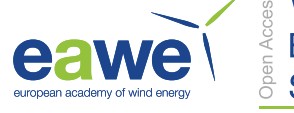

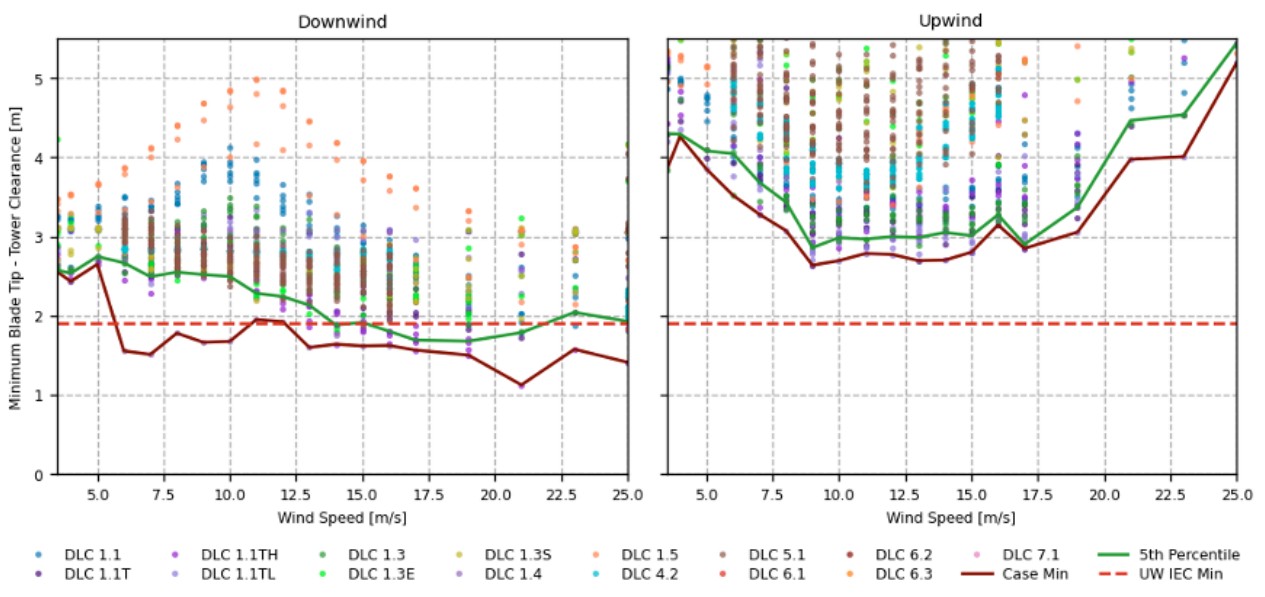

**Figure A5.** Minimum blade–tower clearance between the downwind (left) and upwind (right) configurations. The red dashed line indicates the minimum allowable clearance prescribed by IEC 61400-1 (2019). The experiment clearly violated the minimum clearance and was a source of risk; see Section 3.1.4.

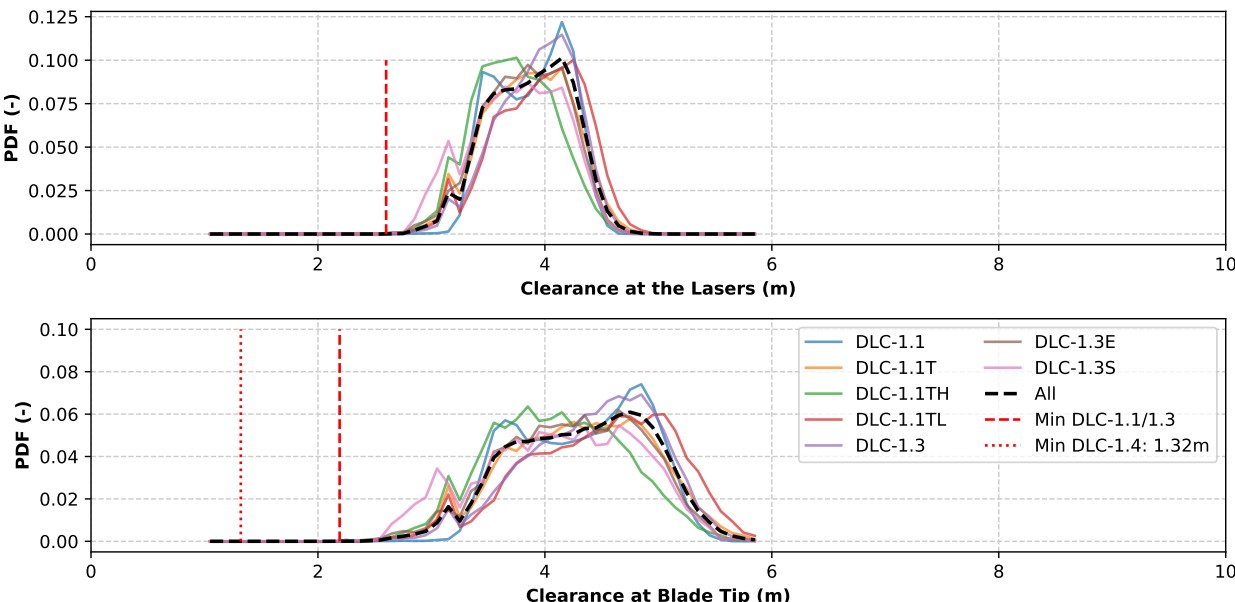

**Figure A6.** Probability density functions (PDFs) of the blade–tower clearance at each blade passage for the six turbulent seeds of each DLC listed in the legend; see Section 3.1.2 for more details.




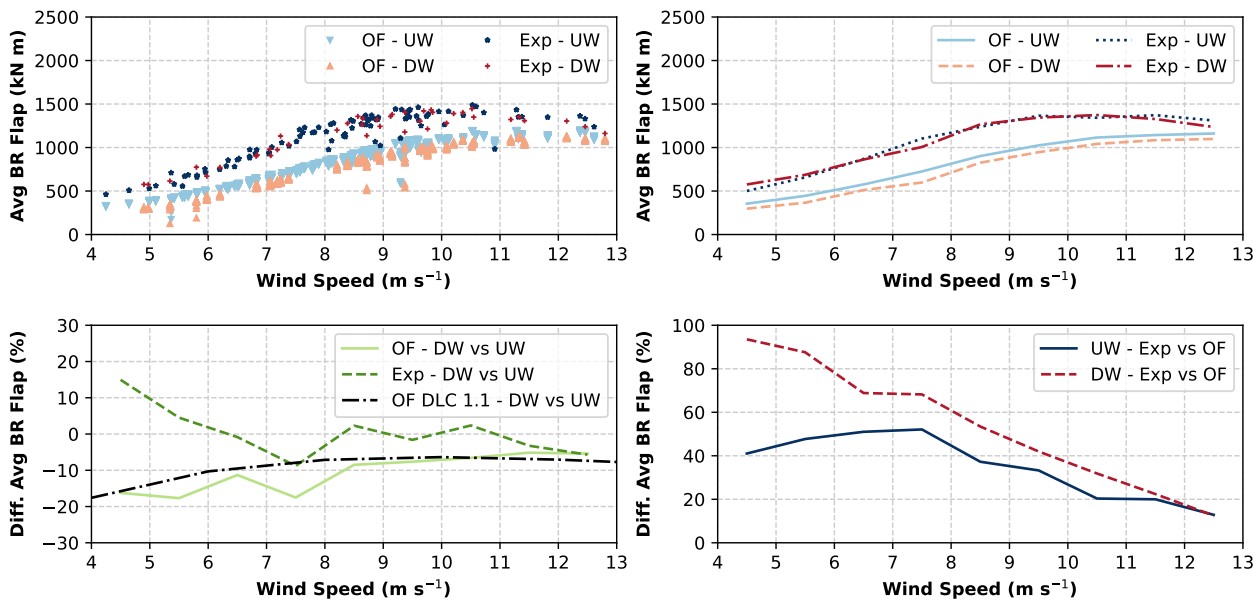

**Figure A7.** Mean blade root (BR) flapwise moment of the upwind (UW) and downwind (DW) configurations in OpenFAST (OF) and in the field (Exp).

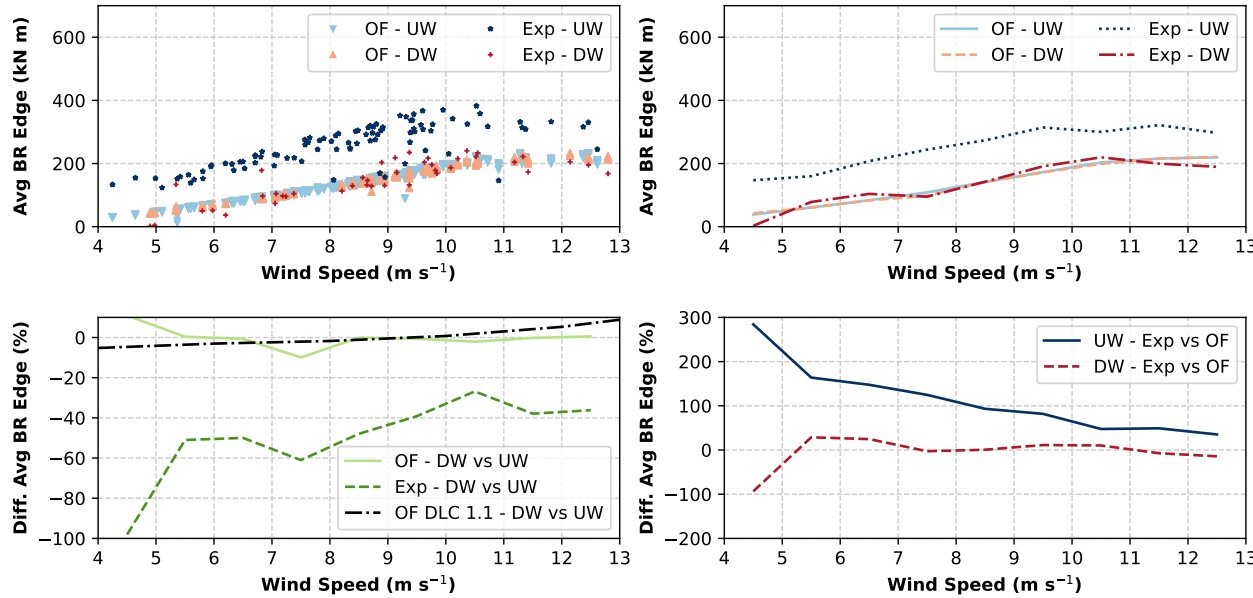

**Figure A8.** Mean blade root (BR) edgewise moment of the upwind (UW) and downwind (DW) configurations in OpenFAST (OF) and in the field (Exp).




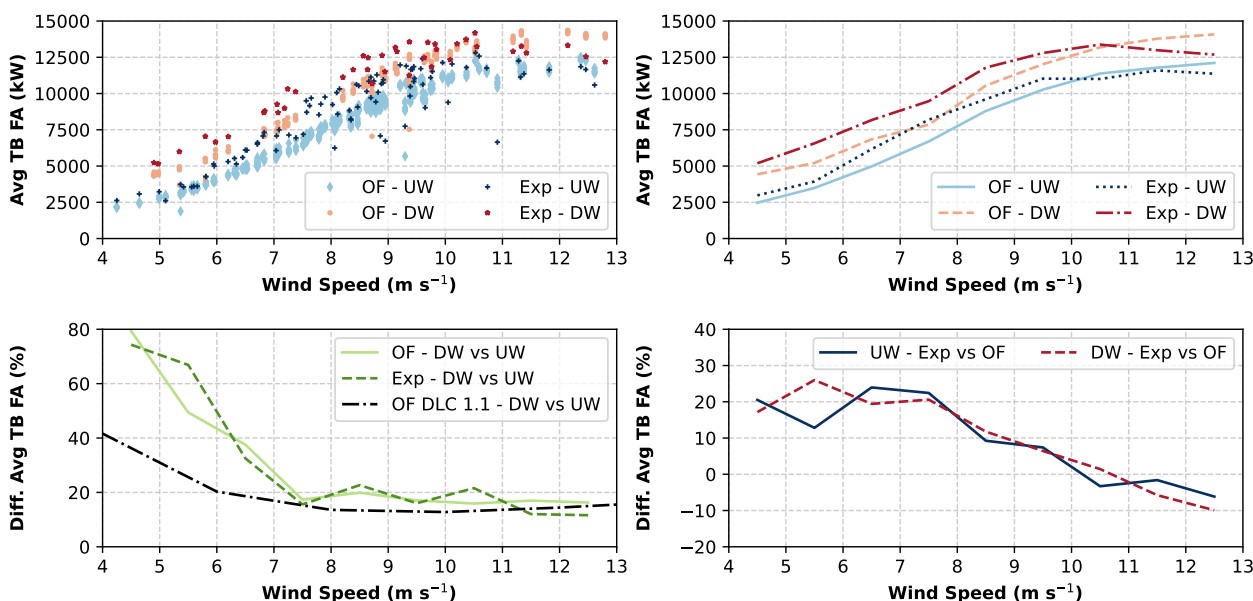

**Figure A9.** Mean tower-base (TB) fore-aft (FA) moment of the upwind (UW) and downwind (DW) configurations in OpenFAST (OF) and in the field (Exp).

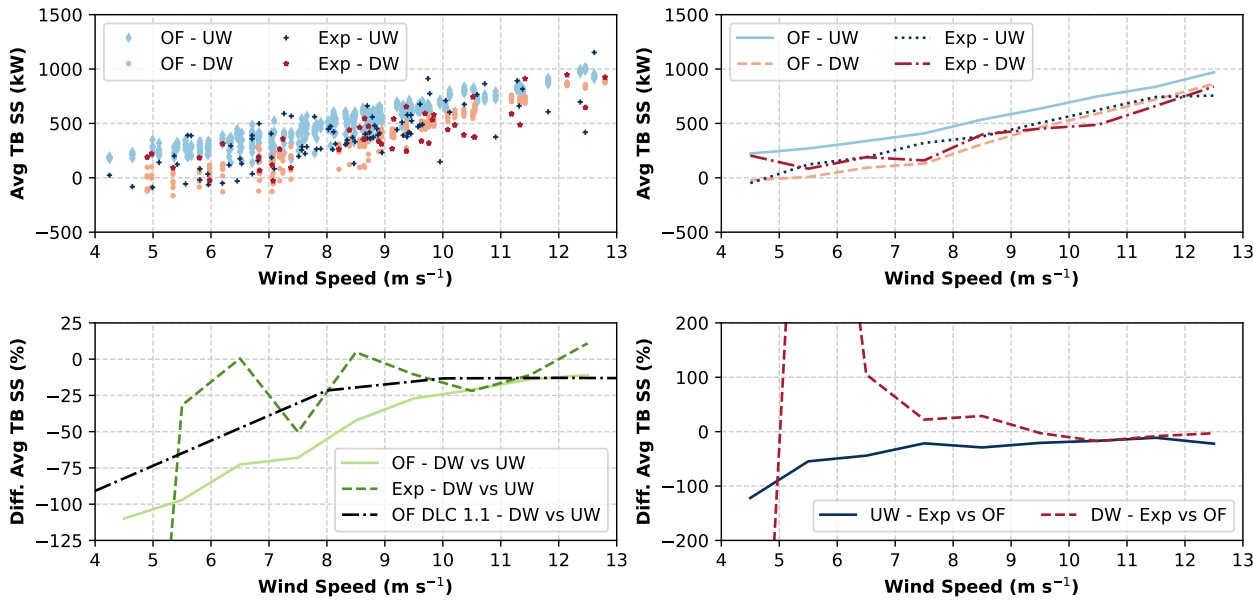

**Figure A10.** Mean tower-base (TB) side-side (SS) moment of the upwind (UW) and downwind (DW) configurations in OpenFAST (OF) and in the field (Exp).

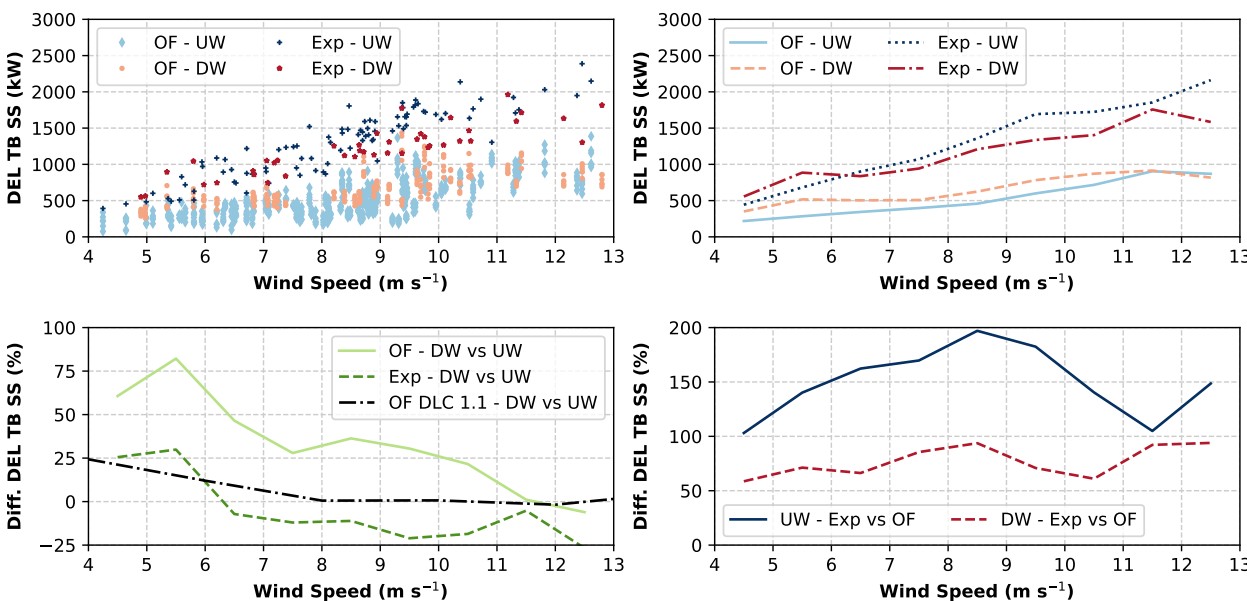

**Figure A11.** DEL of tower-base (TB) side-side (SS) moment of the upwind (UW) and downwind (DW) configurations in OpenFAST (OF) and in the field (Exp).



*Author contributions.* PB led the experiment and the preparation of this manuscript. LJF conducted the photogrammetry to quantify the alignment of the blade pitch and advised during the FMEA. NH characterized the occurrence of extreme events at the NREL Flatirons

Campus. CI led the conversion of the turbine from upwind to downwind, the collection of load data, and the load analysis. MI is the lead technician of the turbine and led all operations on the turbine. JK led the de-risking of the downwind operations on the drivetrain system. SL led the finite element modeling of the main bearing. AH and CW helped with the collection and processing of experimental acoustic data. JR manages the turbine, led the collection and the postprocessing of acoustic data, and co-led the conversion and the operation of the turbine from upwind to downwind. DS led the validation of the OpenFAST model and the load analysis. ST helped with all field operations. All

co-authors contributed to the manuscript.

*Competing interests.* The authors declare that they have no competing interests.

*Acknowledgements.* The support of management at the National Renewable Energy Laboratory and at the Wind Energy Technologies Office of the U.S. Department of Energy was critical in the de-risking and execution of such an ambitious experiment. The support of experts at component manufacturers SKF, Winergy, and Hydac, at GE, and at Gulf Wind Technology was also critical to the success of the experiment.

The assistance of Tom Levet and Robin Woodward from Hayes McKenzie was crucial to achieving the amplitude modulation results. The thorough review of Paul Veers greatly improved the final draft of this manuscript. All these contributions are gratefully acknowledged.

The research was performed using computational resources sponsored by the U.S. Department of Energy's Office of Energy Efficiency and Renewable Energy and located at the National Renewable Energy Laboratory. This work was authored by the National Renewable Energy Laboratory, operated by Alliance for Sustainable Energy, LLC, for the U.S. Department of Energy (DOE) under Contract No. DE-

AC36-08GO28308. Funding provided by the U.S. Department of Energy Office of Energy Efficiency and Renewable Energy Wind Energy Technologies Office. The views expressed in the article do not necessarily represent the views of the DOE or the U.S. Government. The U.S. Government retains and the publisher, by accepting the article for publication, acknowledges that the U.S. Government retains a nonexclusive, paid-up, irrevocable, worldwide license to publish or reproduce the published form of this work, or allow others to do so, for U.S. Government purposes.





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
