# Peer review of "Upwind vs downwind: Loads and acoustics of a 1.5 MW wind turbine"

_Wind Energy Science, 2025_

## Author Comment (AC1)

**REVISION TO MANUSCRIPT DRAFT**

**Journal** Wind Energy Science

**Manuscript ID** WES-2025-8

**Title** "Upwind vs downwind: Loads and acoustics of a 1.5 MW wind turbine"

The authors would like to thank the associate editor Jonathan Whale and the two anonymous reviewers for their time and valuable feedback. Their inputs contribute to the improvement of the paper. A list of point-by-point replies to the reviewers' comments is reported in the following. The text from the reviewers is reported in black. Each point is followed by the authors' reply colored in blue.

**Anonymous Referee #1**

Dear Authors,

I have reviewed your article and would like to share my feedback.

The article presents an in-field test campaign on a 1.5 MW wind turbine that was converted from an upwind to a downwind rotor configuration. Downwind rotors have not been extensively studied. However, as the authors rightly point out, they could play an important role in accommodating highly flexible blades in future turbine designs.

The article offers several contributions that are valuable to the wind energy research community. These include the safety checks and planning required for converting the rotor configuration, the measurement and comparison of performance, loads, and noise for the downwind and upwind configurations, and the validation of an aero-servo-elastic model against field measurements for both configurations.

The research objectives and hypotheses are clearly presented, and the discussion of the methodology and results is sufficiently detailed. The paper is generally well-structured.

For these reasons, I believe the article merits publication in Wind Energy Science. However, I recommend that the authors address the following comments to strengthen the manuscript.

One major point: while the results are interesting and valuable, they are mostly presented without interpretation. The paper would benefit from a discussion that attempts to explain the trends or, at the very least, provides guidance on how to interpret the results.

We would like to thank reviewer #1 for their time and input. The sources of differences between numerical predictions and experimental measurements are not obvious. Nonetheless, we have added some sentences to section 5.1 providing some additional interpretation to the results.

**Specific comments**

- Abstract: The final sentence is overly specific. I recommend focusing instead on the broader potential of downwind rotor technology for future wind turbines. References to floating wind should be avoided, as this topic is not addressed in the article.

We removed the recommendation to investigate the potential of downwind rotor technology for floating wind applications from the abstract. We chose not to elaborate about the broader potential of downwind rotor technology, which should be the result of technoeconomic analyses that are not part of this study.

- R51: "and increased vertical entrainment allows downstream turbines to increase their power output." Please explain what "vertical entrainment" refers to and why it increases when the wake is deflected downward.

We reformulated the sentence.

- R78: "The rotor, however, spins in the opposite direction with respect to the nacelle." Could you briefly clarify whether this has any implications for the wind turbine generator?

As discussed in section 3, the downwind operations did cause a flip in the rotation of the drivetrain components. We've clarified this here early so that readers are not left guessing. Note that other than the swapping of phases due to the reverse rotation, no further actions were needed for the generator. We did also evaluate the risk of generator overheating due to reversed air scoop direction as described in Table A4, but the temperatures are monitored by the SCADA and no issues were encountered during the experiment.

- R79: "the pitch actuators need to operate between 180° and 270°." Please note here that this required a change to the blade home position and some physical modifications to turbine components. Clarify that this is not a problem for the pitch actuators.

These are discussed in Section 3. We added a sentence pointing readers to Section 3.

- R86: Please remove "The next sections describe the design, planning, execution, and results of the experiment."

Removed and replaced with "The structure of the paper is as follows".

- Table 1: Some details, such as the wind turbine manufacturer's address or the turbine serial number, may be unnecessary for understanding the article. Please review and remove any unnecessary information.

We removed the address of the manufacturer and the serial numbers of the main components.

- R142: "reversed aerodynamic thrust". Clarify that in the downwind configuration, the thrust force during normal operation is directed outward from the nacelle, pulling the transmission rather than pushing it.

Done.

- R168: "that was generated with a 3D scanner." Please add the accuracy specification of the scanner.

Unfortunately, we don't have this information available as this analysis was performed by a third-party years ago. This said, we do not think that this is a critical element to include in this article.

- R175: "The measurements show a range of 0.5° and a dependency on the yaw angle, which might suggest that the tower is itself not perfectly vertical." Could this also be due to tower bending from gravitational loads?

No. When yawing the rotor-nacelle assembly the deflections due to gravity should be the same regardless of the yaw angle.

- R195: "ultimate thrust of 250 kN." How was this value determined?

From the load analysis conducted in OpenFAST as discussed in Section 3.1.2. We added a short sentence explaining it.

- R241: "with a photogrammetry process relying on photos shot from the ground while pointing vertically up" Please comment on the accuracy of this process in estimating pitch angles.

We added this sentence to the manuscript "The accuracy of the analysis based on photogrammetry is estimated to be +-0.02 degrees based on the repeatability of the point placement and the resolution of the pictures."

- R245: "to avoid the risk of falling into the deadband of the vane instrument." This is unclear. Please briefly explain what the risk is.

We simply removed the sentence, which is not at all critical and only confuses readers.

- R249–252: This section is not clear. Did you modify the generator, the gearbox, or both?

We had to swap the phases of the generator. We simplified the paragraph to make it more readable and we added some more details about the steps taken to enable the reversed rotor speed.

- R293: "only the day of 13 April 2024…construction being stopped on a Saturday". Earlier it's mentioned that construction was postponed. Please clarify this apparent contradiction.

Thank you for highlighting this inconsistency. During weekdays, we did our best to postpone construction activities. However, the best acoustic measurements were recorded on Saturday April 13. We rewrote a couple of sentences to improve the narrative.

- R315: "average wind speed, average turbulence intensity" Are these values measured at hub height? Were they recorded by the wind vane?

Measured on the met mast at hub height. We've clarified this.

- R366: "Note also that the OpenFAST data refer to the simulations that model the inflow recorded in the field". In the results section, please clarify that OpenFAST simulations used inflow conditions recorded in the field (for model validation) and also tested identical inflow for both rotor configurations (to isolate configuration effects). This dual approach should be emphasized.

Added an extra sentence to emphasize the difference between the two datasets run in OpenFAST.

- R370: "oscillate around the 0% line". Note that this applies only for wind speeds greater than 8 m/s.

We'd say above 7 m/s rather than 8 m/s, but yes, we added this point.

- R374: "This prediction" Specify whether this refers to the small drop or the higher power.

Changed to "The prediction of drop in power".

- R379: "underpredicting this quantity by as much as 100 %" Could you discuss why the downwind configuration shows higher power, why OpenFAST underpredicts it, and why the uncertainty is larger in the downwind case?

Following your first recommendation, we've added some sentence discussing the trends.

- R382–383: "The prediction of increasing DEL matches with both the inflow from the field and the inflow from DLC-1.1". This agreement is especially evident at wind speeds above 7 m/s, where data is more abundant—please specify this.

Done

- R424: "The downwind dataset is compared to the upwind dataset collected during an IEC noise test…" This is unclear. It may be due to the dataset not being clearly marked in the figure. Consider clarifying or labeling the dataset better.

We reformulated the sentence to improve clarity.

- R447: "When the turbine is operating at a rated speed of 18.3 rpm". Consider adding rotor speed (rpm) as a secondary y-axis in the corresponding figure.

We added to the main text and to the caption of Figure 13 that 18.3 rpm = 0.305 Hz and that 3P = 0.915 Hz.

- R451: "which corresponds to a rotor speed below the minimum operational rotor speed of the turbine". Is the turbine fully or partially shut down in this condition? Please clarify.

The turbine cuts in at 11 rpm, whose 3P is 0.55 Hz. Rotor speeds of 8rpm do happen during start up though. We think that the text is good as is.

**Technical corrections**

- The paper refers to "the team" throughout. While not incorrect, scientific writing is typically impersonal. If this style was not a deliberate choice, consider revising for a more standard scientific tone.

Thank you for the suggestion, we changed the text and minimized the use of the wording "the team".

- Figure 2: Move details from the caption (e.g., blue dot location, microphone positions, meteorological tower) to the legend.

We added the legend to the plot.

- Figure 5: Clarify in the caption that the numbers in the matrices represent the count of 10-minute periods.

Done.

- R307: Remove the word "loads".

Good catch, done.

- R320: Replace ", and" with a period.

", and" is correct, no change.

- R365: Confirm whether "binned" or "raw" is correct.

Binned is correct, no change.

- R367: Correct "show" to "shows".

Thank you, fixed.

- R372: Change "numerically" to "in OpenFAST simulations".

Done

- R437: Replace "audible" with "present".

Done.

- R447: Change "rotor speed of the turbine" to "blade passing frequency".

Changed to "3P rotor harmonic".

- R460: Change "+0.4%" to "0.4%".

Great catch, thank you, fixed.

**Anonymous Referee #2**

Dear Authors,

I have reviewed the article and summarized my feedback below.

The article has clear research objectives, well-presented hypotheses, and a detailed discussion of methodology and results. Its overall structure is good.

The article contributes to the wind energy research community. It explains the procedure of operating an upwind rotor in a downwind configuration and gives the details and analysis of the possible risks. Performing both the measurements and numerical simulations helps compare the downwind and upwind cases as well as validate the numerical method.

My suggestions for improving the manuscript are listed below.

**General comments:**

- The term "team" is used frequently, which is not common for a technical article.

Fixed (reviewer #1 also recommended this change)

- Too much detail is given about the turbine and its manufacturer.

Fixed (reviewer #1 also recommended this change)

- The first sentence of the abstract states: "This paper discusses the motivation, preparation, risk mitigation, execution, and results of a full-scale experiment where the rotor of a 1.5 MW wind turbine was operated in a downwind configuration" It would be more clear if it is mentioned that the 1.5 MW wind turbine is an upwind wind turbine.

Fixed.

- Lines 56, 183, ..: +40%, +20%, ...: + sign can be left out.

Thank you for the suggestion. The communications department of NREL has performed a professional editing step of the text before submission and recommended this format. The proofreading stage will also provide its recommendations, and we will follow them.

- Line 77: "For the same wind and the same observer, the rotor keeps spinning clockwise when viewed from upwind. The rotor, however, spins in the opposite direction with respect to the nacelle." Does it mean "the same wind direction and observer position" here? Also, the second sentence is not clear.

We followed your recommendation, and we switched to "For the same wind direction and the same observer position, …". We also improved the second sentence as recommended by both reviewers.

- Line 100: "During the experiment, the team made sure to minimize sources of background noise." Although the details are given later, this statement sounds subjective in its own. Please add here that the details of these efforts are given below.

This sentence was flagged by both reviewers, and we changed it pointing the user to Sections 4.2 and 5.2.

Line 121 and caption of Figure 2: Microphone locations are explained in the text and in the caption of the figure. First of all, the detailed explanations in the caption should be moved to the text. Secondly, these measures should be given in rotor diameter (D).

We added the legend to Figure 2, which should now be easier to read.

- Line 168: "...3D computer-aided design model that was generated with a 3D scanner". It is not clear what kind of design model was used and what was exactly generated. Could you please elaborate on this 3D design model?

Unfortunately, this study was performed by a third-party years ago and we do not have access to any detail. As replied to reviewer #1, the 3D model obtained from the 3D scan was only used for an additional check on the accuracy of the model and we do not think that any more detail is critical to this article.

- Line 170: "However, the masses listed in the technical documentation include the fixture used for shipping..." Isn't it possible to find out the (approximate) weight of the mentioned fixture.

Unfortunately, not. The turbine was installed in 2008, and we do not have any detail about these fixtures.

- Table 2: Does the "Blade" refer to "Blade mass"?

Thanks for catching this inconsistency. "Mass" was in the header of the table but also in some of the rows. We fixed the draft leaving the word "Mass" only in the header.

- Line 191: "In upwind operations, thrust at the main bearing is received by a shoulder in the housing and then transmitted to the bedplate. There is no shoulder, however, in the upwind direction." Shouldn't the last sentence refer to "downwind direction"?

No, upwind direction. Regardless, we improved the readability of this sentence distinguishing between "nacelle side" and "rotor side".

- Line 195: "The value was set 50 kN higher than the ultimate thrust of 250 kN.". How is the ultimate thrust determined?

See reply to reviewer #1. We added the sentence "from the DLCs run in OpenFAST as discussed in Section 3.1.2".

- Line 205: "Converting a downwind rotor from upwind to downwind...". The second word, "downwind", seems unnecessary.

Good suggestion, we removed the first "downwind".

- Line 242: "Table 3 lists the three pitch angles for both upwind and downwind operations, reconstructed via photogrammetry." Please indicate the accuracy of these measurements.

The issue of accuracy of photogrammetry was also highlighted by reviewer #1. We added this sentence to the manuscript "The accuracy of the analysis based on photogrammetry is estimated to be +-0.02 degrees based on the repeatability of the point placement and the resolution of the pictures."

- Line 315: "...the average wind speed, average turbulence intensity, and average exponential shear exponent." Please explain how these data are averaged.

This is the simple average for each 10 min data point. We added this wording to the draft.

- Line 370: "... both OpenFAST predictions and experimental recordings oscillate around the 0 % line ..." This is not the case for wind speeds below 7m/s.

This issue was also flagged by reviewer #1 and we added the wording "above 7 m/s".

Lines 414-418: From the results presented in section 5.1, it is clear that OpenFAST is not as accurate in predicting the downstream configuration as in the upstream cases. Could you please elaborate on the possible reasons for this difference in predictions?

This is an interesting point, and we added the following text to the manuscript. "In some metrics, OpenFAST seems to be doing better at predicting quantities for the upwind rotor than for the downwind rotor, namely generator power, blade root edgewise moment DEL, and possibly tower base fore-aft moment DEL. This is however not the case for blade root flapwise moment DEL, nor for many of the quantities plotted in the appendix, namely mean blade root edgewise moment, mean tower base fore-aft moment, mean tower base side-side moment, and tower base side-side moment DEL. In Madsen et al. the accuracy of the tower shadow model is being investigated thanks to measurements recorded by pressure belts. Those findings will shed more light into the accuracy of aeroelastic models for downwind rotors."

Figure 13: Does this figure refer to a single rotor speed (18.3 rpm) as indicated in text line 448? Please make this clear.

Reviewer #1 also flagged this weakness of the previous draft. We have now improved the paragraph.